



# Human-induced influence on eggs and larval fish transport in a subtropical estuary
**Maria Helena P. António[a,c,@], José H. Muelbert[b] and Elisa H. L.Fernande[a]**
[a]Laboratório de Oceanografia Costeira e Estuarina. Instituto de Oceanografia. Universidade Federal do Rio Grande,
Brasil
[b]Laboratório de Ecologia do Ictioplâncton, Instituto de Oceanografia. Universidade Federal do Rio Grande, Brasil
[c]Escola Superior de Ciências Marinhas e Costeiras. Universidade Eduardo Mondlane, Moçambique
[@]E-mail: mhbeula2@gmail.com
## Abstract

9        The transport during the early stages of life to the nursery areas is one of the main processes
in the maintenance of the marine fish population, and human interventions can interfere with this
process. In this work, the TELEMAC-3D model coupled to passive particles was used to understand
the effect of the change in the configuration of the Barra Jetties of the Rio Grande regarding the
transport of eggs and larvae of the croaker *Micropogonias furnieri* in the Patos Lagoon estuary
(PLE). Twelve experiments of 5 days that consisted of periods of high and low discharge combined
with winds from the south quadrant (SW, S, and SE) were carried out to test the hypothesis that
human interventions in the coastal region alter the transport patterns of fish eggs and larvae. The
low flow guaranteed a greater extent of saline intrusion and larvae incursion in the estuary, with the
opposite occurring in the scenario of high flow. The SW wind ensured the most efficient recruitment
into the estuary, in terms of both entry time and maximum reach in both configurations. However,
the recent modernization works of the Barra Jetties have changed the pattern of transport and
dispersal of larvae and have reduced the amount and reach of the incursion of croaker eggs and
larvae into the estuary compared to their old configuration. With the new configuration of jetties,
reductions in the larvae concentration and abundance in the estuary were registered at
approximately 25% for SW and S winds, 68.6% for SE wind at high discharge, and 0.5% to 1% for
winds at low discharge. The lateral stratification in the access channel to the estuary, an important
parameter in the larvae transport and distribution between the jetties and the predominant wind
direction, was decisive in defining the initiation time of the stratification. With the old
configuration, the lateral stratification was established 1 h, 7 h, and 10 h after starting the simulation
with the incidence of SW, S and SE winds, respectively. In the new configuration, the lateral
stratification was established at the same time only with the SW wind, but with a reduced salinity
gradient. In this configuration, only the beginning of stratification was observed at the estuary
mouth with S winds, while the stratification was not established with SE winds. This fact influenced
the intrusion of saline water and resulted in a smaller number of larvae between the jetties and
consequently their transport into the estuary. With the new configuration, a reduction in the
maximum penetration of the larvae within the estuary was observed at 1.6 km for high discharge
and 2.3 km for low discharge. Despite limitations inherent to the numerical modeling technique, the
results obtained corroborate the hypothesis that human interventions in the coastal region change
the patterns of transport of fish eggs and larvae. Furthermore, the findings suggest that
modernization works of the jetties have contributed to reducing the transport of dependent estuarine
species to the Patos Lagoon estuary. Coupled with the knowledge obtained by other research about
this species, this knowledge can support provisioning measures for better management of fishery
resources in the region.
**Keywords:** *Micropogonias furnieri*, larvae transport, anthropogenic effects, ports, Patos Lagoon,
TELEMAC-3D



# 1. INTRODUCTION

Coastal and estuarine environments are extremely important for the life cycle of various marine organisms (Muelbert and Weiss, 1991; Able, 2005; Whitfield, 2016), which are the most important aquatic resources in the world (Liu and Chan, 2016). In addition to their ecological wealth and capability of providing high rates of primary production and abundance of food, estuaries also serve as a habitat, nursery and protection against predators in early life stages and facilitate the development of numerous marine species (Liu and Chan, 2016; Teodosio et al., 2016).

The planktonic phase of species demands attention as a research topic, as it is a characteristic component of the life cycle of most marine organisms (Tiessen et al., 2013), and in fish, it is marked by the stages of eggs and larvae. Changes in transport during these stages have been suggested as one of the important factors that affect the variability of recruitment in marine fish stocks (Brown et al., 2000; Houde, 2008). The survival of fish larvae depends on the pattern of circulation and transport from the spawning to the nursery area, and local biological and physical events can explain the patterns of growth and survival in the early stage of fish life (Lough et al., 1994; Brown et al., 2000; Hinrichsen, 2009; Endo et al., 2019).

Studies to understand the pattern of transport and dispersion of eggs and larvae in estuaries and coastal regions in their passive phase have been of great importance for understanding the reduction in the stock of adults in different regions around the world. Lough et al. (1994) evaluated the influence of advection promoted by the wind in the interannual variability and distribution of cod eggs and larvae in the Georges Bank region. Blanton et al. (1999) reported the use of passive larvae of white shrimp and blue crab megelope to understand the response of the wind transport generation over the shallow estuary channel in the southeastern United States.

Brown et al. (2000) investigated the importance of tidal forces in the transport of larvae through the narrow channel under the effect of jetties in the bay of Aransas Pass. Sentchev and Korotenko (2007) evaluated the effect of physical forcing and vertical migration on the transport and dispersion of sole larvae in the region of freshwater influence (ROFI) in the east of the English Channel. Tiessen et al. (2013) determined the importance of passive transport by advection in the dispersion of eggs and larvae of plaice fish in the south of the North Sea and the English Channel. Teodosio et al. (2016) described the biophysical processes involved in the recruitment of fish larvae in the Ria Formosa lagoon estuary. Joyeux (2001) evaluated the influence of wind, tide, and meteorological forces in the retention of fish larvae transported to estuaries by the Beaufort channel in North Carolina.

The croaker *Micropogonias furnieri* is one of the most important fishery resources in Brazil, with approximately 43.369 tons of catch per year (MPA, 2011). Along the coast of Rio Grande do Sul, in the Patos Lagoon estuary, *M. furnieri* is one of the 3 most abundant species of Sciaenidae (Ibagy and Sinque, 1995) and is considered one of the species with the greatest commercial value in the region (Haimovici and Cardoso, 2017). The species is in decline in the Patos Lagoon region and its catch was reduced from 22,500 tons between 1970-1974 to 7,000 tons captured between 2007-2010 (Haimovici and Cardoso, 2017). Croaker spawns preferentially along the internal continental shelf, near estuarine mouth systems, and can spawn an average of 3 to 7 million eggs over a breeding season (Acha et al., 2008; Alburquerque, 2008;Alburquerque et al., 2009; Bruno and Muelbert, 2009; Acha et al., 2012). It is present throughout the year, and the summer (November to April) is the period of its greatest spawning, consequently of eggs and larvae abundance (Muelbert and Weiss, 1991; Ibagy and Sinque, 1995). In the South Atlantic, the physical mechanisms that determine the success of the transport and recruitment of the early stages of the life of the croaker are consensual and attributed to winds and low river discharge (Muelbert and Weiss, 1991; Ibagy and Sinque, 1995; Acha et al., 1999; Martins et al., 2007; Acha et al., 2012; Costa et al., 2014; Franzen et al., 2019).

The Patos Lagoon estuary is known to have favorable conditions for feeding and development for numerous species, both fish and crustaceans (D'Incao, 1991; Salvador and



Muelbert, 2019). In this estuary, the dynamics of input and output of estuarine-dependent organisms
are controlled by winds and discharges because these are the dominant forces in circulation (Moller
et al. 2001; 2009). The recruitment of eggs and larvae in the Patos Lagoon estuary occurs during the
spring and summer (Ibagy and Sinque, 1995; Martins et al., 2007; Vaz et al. 2007; Bruno and
Muelbert, 2009; Franzen et al., 2019), and the mechanism responsible for its entry into the estuary
is the sea level rise on the coast that is generated by the Ekman transport resulting from the south
quadrant winds (Vaz et al. 2007). Vaz et al. (2007) highlight that the residual baroclinic current
resulting from the contribution of the continental discharge promotes the retention and
accumulation of the ichthyoplankton inside the estuary. The dynamics of transport and dispersion of
eggs and larvae of the *Micropogonias furnieri* in the Patos Lagoon estuary are determined by
weather conditions, such as the direction, intensity and duration of the southwest winds, combined
with low discharges, passing each stage of their development in different environments of the
estuary (Martins et al., 2007; Bruno and Muelbert, 2009; Franzen et al., 2019).
Changes in the topography and natural geomorphological structure of the access channels
impact the dynamics and ecology of estuaries (Yuk and Aoki, 2007; Liu and Chan, 2016).
Specifically, concerning the transport of fish eggs and larvae, studies show that there is a direct
correlation between recruitment and environmental conditions during their transport, making
anthropogenic contribution one of the processes that limit stock and recruitment (Hinrichsen, 2009;
Acha et al., 2012). In 2010, modernization works were completed at the mouth of the Patos Lagoon
estuary (Moller and Fernandes, 2010). The Barra Jetties, built-in 1915, had an increase in the length
of approximately 10% and 18% (370 m and 700 m) on the east and west side, respectively, as well
as a reduction of approximately 15% in the opening width (currently with 700 m). An increase in
the depth in the navigation channel was followed by a reduction in saline intrusion and the current
speed that was associated with the modifications made (Lisboa and Fernandes, 2015; Silva et al.,
2015). Recently, António et al (submitted) found that changes in the access channel and Barra
Jetties caused a reduction in saline intrusion, a reduction of approximately 20% in both flooding and
ebbing speeds, and a reduction in the time of occurrence of lateral stratification events between the
jetties by approximately 1/3 from the old to the new jetty configuration. These changes in the
characteristics of estuarine circulation may have an impact on the transport and dispersion of eggs
and fish larvae in the PLE.
The high cost and the difficulty in obtaining data in situ with adequate space-time resolution
for analyzing the complexity of coastal ecosystems have limited its studies. The numerical
modeling technique, coupling hydrodynamic and biological models, has been increasingly used as a
tool to solve this limitation ( Lough et al., 1994; Brown et al. 2000; Seiler et al., 2015;Franzen et al.,
2019). The physical-biological coupling has ensured better coverage by interpolation and
extrapolation of data in the space-time domain, assisting in fish dynamics studies at an early stage
of life. These advances have enabled us to understand the causes of mortality of larval and juvenile
fish during transport, focusing on the effects of advective and tropodynamic processes ( Brown et
al. 2000; George et al, 2011; Seiler et al., 2015). Lagrangian models of particle transport consider
eggs and larvae to be passive particles, allowing the monitoring of their trajectory from the
spawning site to their final deposition (Blanton et al., 1999; Brown et al. 2000; Martins et al., 2007;
Vaz et al. 2007; Acha et al., 2012).
The main objective of the present study is to determine if the modification of the Barra
Jetties of the Rio Grande influences the transport of eggs and larvae of the croaker, *Micropogonias
furnieri,* in the Patos Lagoon estuary. For this, the hydrodynamic model TELEMAC-3D with
passive particles will be used. The results of this study aim to contribute to an understanding of the
new dynamics of the recruitment process and management of fishery resources of the Patos Lagoon
and the adjacent coastal region.
**1.1. Study Area**





The Patos Lagoon (Figure 1) is located in the southwestern region of Brazil between 30º and
32º South. It is classified as a strangled coastal lagoon (Kjerfev, 1986) that is 250 km long, 40 km
wide, and an average depth of 5 m, occupying an area of approximately 10,360 km$^2$ (Moller et al.,
2001). The lagoon is connected to the South Atlantic Ocean by a narrow channel less than 1 km
wide (Martins et al., 2007). The estuarine region of the Patos Lagoon, which represents
approximately 10% of the total area of the lagoon, has more than 80% of its area with depths below
2 m and has a diversity and abundance of flora and fauna, which makes these local areas suitable
for the development of estuarine-dependent organisms (Moller et al., 2001; Odebrech et al., 2010).
The lagoon has 3 main tributaries, the Guaíba River, the Camaquã River, and the São
Gonçalo Channel, with an average discharge of approximately 2400 m$^3$s$^{-1}$, ranging between 700
m$^3$s$^{-1}$ during the summer and 3000 m$^3$s$^{-1}$ during spring (Moller et al., 2001; Moller and Fernandes,
2010). Tides have little influence on the estuary dynamic (which is defined by winds and
discharges), presenting a diurnal predominance with an amplitude of approximately 0.3 m and is
attenuated during the propagation toward the estuary (Moller et al., 2001; Fernandes et al ., 2004;
Moller et al., 2009). Its greatest contribution is in modulating the mixture of the water column and
transporting water further to the north of the estuary during periods of less intense winds and
discharges (Moller and Fernandes, 2010).
The winds and discharge regime delimit the range of saline intrusion in the Patos Lagoon,
and in low discharges, the salinity can pass the northern limit of the estuarine region (Moller et al.,
2001; Moller and Fernandes, 2010; Seiler et al., 2015). In contrast, the high discharges function as
physical barriers, not allowing the intrusion of saltwater into the estuary, which may affect the
pattern of recruitment, immigration, and emigration of organisms of estuarine species (Garcia et al.,
2001; Salvador and Muelbert, 2019).

## 2. METHODOLOGY

The present study was based on the application of the hydrodynamic numerical model
TELEMAC-3D (www.opentelemac.org) and its Lagrangian module to investigate the influence of
configuration change of the Barra Jetties of the Rio Grande on the transport and dispersion of eggs
and larvae of the croaker, *Micropogonias furnieri*, in the Patos Lagoon estuary. Controlled
simulations were carried out, considering extreme discharge conditions and winds from the south
quadrant.

### 2.1. Hydrodynamic Numerical Model

The TELEMAC-MASCARET model (V7P0 version) was developed by *Laboratoire*
*National d'Hydraulique et Environnement of the Company Electricité of France (©EDF)*. The
model presents modules in two and three dimensions to study hydrodynamics, sediment transport,
waves, and water quality of coastal regions. The hydrodynamic model solves the Navier-Stokes
Equations, considering local variations of the free surface of the fluid, neglecting the variation of
density in the mass conservation equation, considering the hydrostatic or non-hydrostatic pressure
and the Boussinesq approximation to solve the equation of motion. The model applies the Finite
Element Method in order to solve the hydrodynamic equations, using the Sigma Coordinate System
for vertical discretization. The model domain is discretized by a non-structured grid of finite
elements (triangular elements), which allows concentrating a higher number of elements in regions
of interest and/or significant bathymetric variations, and lower resolution in regions of more
homogeneous bathymetry, reducing computational time. Details about the model formulations are
presented by Hervouet (2007).
The bathymetry of the Patos Lagoon, the estuary, and the adjacent coastal region was
obtained from historical data. Nautical charts from the Directory of Hydrography and Navigation
(DHN, Brazilian Navy) before 2010 were used as the "old" bathymetric information (before
changes in configuration). Data from the jetty expansion project were used to define the bathymetry





after the alteration of the jetties. The main difference between the two grids is the length of the
jetties and the depth of the access channel to the estuary (Figure 1D and 1E). The BlueKenue
Software was used to generate the unstructured bathymetric grids of triangular elements. Grid
optimization was made in the complex morphology and shallow areas inside the estuary and at the
adjacent coastal region, allowing higher resolution in regions of interest. Two resulting meshes were
used to reproduce the hydrodynamics before and after the modification of the jetties (Figures 1D
and 1E). The meshes encompass the entire study area up to about 2500 m depth to better represent
the coastal dynamics.
The open boundaries of the domain were forced with results from regional and global
models and field data. To be comparable, simulations for both scenarios had the same set-up. Time
series of daily averaged river discharge of the main tributaries (Guaíba river and Camaquã river,
Figure 1) were obtained from the National Water Agency (www.ana.gov.br) and prescribed at the
northern and central continental boundaries. The mean discharge data for the São Gonçalo Channel
was considered constant as 700 m³/s (Vaz et al., 2006), as there were no time series of discharge for
the studied periods. Temperature and salinity fields obtained from the HYCOM model (Hybrid
Model Coordinate Oceanic, https://hycom.org/), with a temporal resolution of 3h and spatial
resolution of 1/12.5°, were prescribed tridimensionally in all grid points. Wind time series, with a
spatial and temporal resolution of 0.75º and 6h, respectively, were obtained from the ECMWF
(European Center for Medium-Range Weather Forecasts, www.ecmwf.int). Eleven (11) sigma
levels were considered in the vertical and distributed from the bottom to the sea surface.
The model calibration and validation for both scenarios (Supplement A) are presented in
more detail by António et al. (2020, submitted). The calculated results were compared with field
data for the period between October and November 2006 for the old jetty configuration and October
to November 2010 for the new configuration. The model performances (Table 1, Supplement A)
ranged from Good to Excellent considering the Root Mean Square Error (RMSE) and the Relative
Mean Absolute Error (RMAE) results (Wastra et al., 2001). Current velocity time series were used
for calibration tests (Figure A1, Supplement A) and salinity, sea surface elevation and current
velocity time series were used for validation tests (Figure A2 and A3, Supplement A) for both
configurations.
**2.2. Particle Tracking Model**
The particle model is a subroutine of the hydrodynamic model TELEMAC-3D, which is
simulated internally at each time step after the hydrodynamic component. Therefore, to reproduce
the transport of eggs and larvae in the passive phase and simulate their dispersion in the Patos
Lagoon estuary, the Lagrangian model was coupled to the hydrodynamic model TELEMAC-3D.
The particle model obtains the Lagrangian information from the Eulerian velocity
information that is calculated by the hydrodynamic model to determine the particle trajectory
caused by the flow at each time step, and the three-dimensional trajectories are computed using the
position information calculated at each step of the time.
The horizontal and vertical advection movement considers that the Euler scheme and the
particle buoyancy are based on the zonal, meridional and vertical components (u, v and w) and is
given by the following expressions:

$$X_i\left(x_0, y_0, z_0\right)^{n+1} = X_i\left(x_0, y_0, z_0\right)^n + \int_{t_0+n\Delta t}^{t_0+(n+1)\Delta t} u_i\left(x_0, y_0 z_0, t\right) dt$$

where $X_i$ is the position, for horizontal ($x_0$, $y_0$) and vertical ($z_0$) position of the particle movment; $u_i$ is
the velocity for a zonal ($x_0$), meridional ($y_0$) and vertical ($z_0$), u, v and w velocity component, respectively; t
is the time and $\Delta t$ is the time step.
The calculation of hydrodynamic components is taken into account in a discrete three-
dimensional way at each point of the numerical grid. During the simulation, the particles are free to





move to any position between the grid points. In each new time step, the velocity is interpolated
instantly for each position where the particles are located. The accuracy and resolution of particle
transport calculations are extremely dependent on hydrodynamic terms.
In this study, a time step of 60 s was applied, with the particle model in phase with the time
step of the hydrodynamic model. Thus, at every 60 seconds, TELEMAC-3D runs the hydrodynamic
component, and the results are instantly inserted into the particle model, with the displacement of
the particles being calculated three-dimensionally in the numerical grid.
**2.3. Model Experiments**
To investigate the effect of the modifications made in the Barra Jetties of the Rio Grande on
the transport and dispersion of eggs and larvae of *Micropogonias furnieri* in the Patos Lagoon
estuary, a total of 12 controlled experiments were carried out, with dynamic forces from extreme
conditions of continental discharge combined with constant south quadrant winds (SW, S, and SE),
considering the old and the new configuration of the Barra Jetties (Table 1).
Table 1: Controlled experiments simulations

| Wind direction | Low discharge (La Niña, 2012) | | High discharge (El Niño, 2003) | |
|---|---|---|---|---|
| | Old configuration | New configuration | Old configuration | New configuration |
| SW | X | X | X | X |
| S | X | X | X | X |
| SE | X | X | X | X |

The simulations were carried out for the first 5 days of January, which represented the
passage of cold fronts and ensured the continuous incidence of winds from the south quadrant in the
region. Extreme high and low discharge regimes for the years 2003 and 2012, respectively, were
considered. These extreme discharge regimes are associated with ENSO, with high discharge
characteristics during El Niño (2003) and low discharges during La Niña (2012) (Moller and
Fernandes, 2010). Constant SW, S, and SE winds were also considered with an initial intensity of 8
m.s$^{-1}$, decreasing linearly after the second day of incidence until reaching 4 m.s$^{-1}$ on day 5. South
quadrant winds were considered to be those that facilitate the entry of saltwater into the estuary
(Moller et al, 2001).
The simulation time of 5 days considered the growth rate of the larvae of the species under
study and their passive period in the plankton. Eggs of the *Micropogonias furnieri* hatch in up to
approximately 24 hours (Albuquerque, 2008), and the larvae are approximately 1.85 mm. The
average growth rate is 0.36 mm/day (Albuquerque et al., 2009), and at the end of 5 days after
spawning, the larvae will be approximately 3.29 mm. The spawning of the particles was done only
once, at the mouth of the Patos Lagoon estuary (Figure 1D and 1E), considering the grouped
spawning characteristic of the species. The spawning site was defined based on past studies by
Martins et al. (2007) and Franzen et al. (2019), who concluded that the spawning at the estuary
mouth guarantees the best recruitment of eggs and larvae of the croaker to the Patos Lagoon
estuary.
Due to the computational limitation of the TELEMAC-3D version V7P0, the maximum
particle concentration per defined spawning area was 7000 for each simulation. This number of
particles represents approximately 25% of the maximum average concentration of eggs per cubic
meter (497 eggs/100 m$^3$) (Bruno and Muelbert, 2009). Then, for each experiment run, 7000 particles
were placed at a depth of 5 m at 00:00 on January 1, 2003 (high discharge) and January 1, 2012
(low discharge). The evolution of the larvae was monitored for 5 days.





Eggs and larvae were considered passive and neutral particles, assuming that fish eggs are
transported by the flow without depositing. During the simulation, the particles were considered
eggs from the spawning site for up to 24 hours, and shortly afterward, the occurrence of hatching of
the eggs was considered when the particles started to be considered larvae of the *Micropogonias*
*furnieri* croaker. The abundance of eggs and larvae in a given region is affected by predators, by the
rate of growth and mortality, but these processes were not considered in the present study due to the
limitations of the Lagrangian model.

### 2.4. Data Processing

Numerical simulations that considered the old and the new configuration of the Barra Jetties
of the Rio Grande and the incidence of constant winds of the south quadrant (Figures 2A - 2F) in
periods of high (Figure 2G) and low (Figure 2H) continental discharge were analyzed
comparatively for the 6 simulated scenarios (Table 1).
The results were analyzed in relation to the extension of entry of larvae and eggs and their
distribution in the estuary. Maps of spatial patterns of the salinity field and the final distribution of
the larvae in the estuary in the last step of the simulations for the periods of high and low discharge
during the incidence of southern winds were selected for presentation.
Based on the sampling techniques proposed by Cochran (1976) and based on simple
significant sampling calculations made by Miaoulis and Michener (1976), stratified random samples
of 99 larvae, which represented a 10% precision level (sampling error), a 95% confidence level and
a variability degree (proportion) $P = 0.5$, were extracted from the total of 7000 placed in the
simulations (Cochran, 1976; Miaoulis and Michener, 1976; Israel, 1992). To determine the average
path taken by the larvae at the end of each day in the two configurations (old and new), the
weighted distances traveled by each of the larvae that compose the sample were calculated from the
spawning place (in the mouth of the estuary) until the end of each of the 5 days of simulation. Then,
the center of mass was found, calculating the average distance covered at the end of each day.
Finally, the mean standard deviation of the individual distances from the center of mass (mean
distance) was calculated. The Student's t-test was applied to verify the significance between the
average distances in the two configurations in the simulated scenarios (Louangrath, 2015;
Padovani, 2012). At random, another reduced sample of 10 larvae (P1, P2, P3, P4, P5, P6, P7, P8,
P9, and P10) was extracted at the end of each day, and the trajectories of larvae were tracked from
the spawning site to the final location in the two configurations (old and new) of the Barra Jetties, at
the end of each of the five simulation days.
The abundance of larvae for each hydrodynamic simulation was extracted from the model
result for 6 areas (A1, A2, A3, A4, A5 and A6 (Figure 1C)) at the end of each of the five simulation
days and analyzed in terms of the distribution of the spatiotemporal concentration along the estuary.
To investigate changes promoted by lateral stratification in the distribution of eggs and
larvae between the Barra Jetties, profiles of the spatial distribution of salinity and eggs and larvae
were extracted between the Barra Jetties during the flood period. Changes in the time of occurrence
of lateral stratification and the salinity gradient were observed in the hydrodynamic study with
dynamic winds (António et al., 2020, submitted). In this way, the aim is to evaluate the effect of the
difference in the direction of the incident wind (SW, S, and SE) on the variability of the behavior of
the lateral stratification, and consequently on the changes in the distribution of eggs and larvae that
occurred due to the recent modernization works.
To associate the direction of the incident wind and the position (west, center channel and
east) of the eggs and larvae among the jetties, the internal area between the west and east Jetties was
divided into 3 (three) regions: root, center and mouth of the jetties (mouth). Each of these areas was
subdivided into 3 other areas (west, center channel and east jetties), totaling 9 areas, where the
concentrations of eggs and larvae were counted during the occurrence of lateral stratification and
incidence of SW, S and SE winds.



## 3. RESULTS

The periods of extreme discharges used in the experiments (Figure 2) presented different
characteristics. During the high discharge period (El Niño, January 2003), the average discharge
was 6340 $m^3s^{-1}$, the maximum was 8000 $m^3s^{-1}$ (on January 2), and the minimum was 5000 $m^3s^{-1}$ (on
January 5). In contrast, in the low discharge period (La Niña, January 2012), it was practically
constant at 1200 $m^3s^{-1}$ over the 5 days.

### 3.1. Saltwater Distribution

During the period of high continental discharge, the penetration of saltwater was relatively
less in the new configuration in the 3 simulated wind scenarios (Figure 3). During SW winds, the
salinity in the new configuration (Figure 3D) reached 54 km, approximately 3 km less than the old
configuration, which reached 57 km (Figure 3A). During the S wind, salinity reached 43 km in the
old configuration (Figure 3B) and 41 km in the new configuration (Figure 6E). During the SE wind,
the salinity reached 36 km and 34 km in the old (Figure 3C) and the new (Figure 3F) configuration,
respectively.
In the low discharge period, the extent of saltwater intrusion passed from the northern limit
of the Patos Lagoon estuary in all simulations (Figure 4). The results indicate that the lagoon was
less saline in the new configuration in relation to the old configuration during the SW wind, and
during the S and SE wind, the saltwater intrusions did not present any noticeable differences. At the
end of the 5 days of simulation, the 5 psu isohaline reached approximately 106 km in length in the
old configuration (Figure 4A) during the incidence of the SW wind. In the new configuration, the
saline intrusion was reduced to 97 km (Figure 4D). During the S wind, the saline intrusion had an
extension of approximately 96 km both in the old (Figure 4B) and in the new configuration (Figure
4E). The SE wind was the one that presented the least saline intrusion. The salinity for the old
(Figure 4C) and new (Figure 4F) configuration reached approximately 79 km.

### 3.2. Transport and Dispersion of Larvae

A reduction in the extent of larval transport after the modernization works was observed in
the 3 simulated wind scenarios for the period of high (Figure 5) and low continental discharge
(Figure 6). The SW winds were the ones that guaranteed the largest incursion of the larvae in the
high and low discharge, both for the old configuration (Figures 5A and 6A) and the new
configuration (Figures 5D and 6D).
At high discharge, during the SW wind, the larvae extension was approximately 58 km and
56 km for the old and new configuration, respectively (Figure 5A and 5D). The S and SE winds did
not show great differences in their extension between the old and the new configuration. The length
varied between 42 km and 40 km for the S wind (Figure 5B and 5E) and between 38 km and 37 km
for the SE wind (Figure 5C and 5F) for the old and new configuration, respectively. Differences in
the final location of the larvae were observed that were associated with the direction of the incident
wind because, during the SW and S wind, the larvae were closer to the east side of the lagoon, while
the opposite was observed during the SE wind incidence.
Differences in the extension of larvae penetration were noticeable mainly in low discharge,
with their pattern associated with the variability of the direction and intensity of the incident wind
(Figure 6). The highest concentrations were observed along the navigation channel, with few larvae
entering the shallow region of the bags (Arraial and Mangueira, Figure 1). The larvae showed a
greater extension than that defined by the 5-psu salinity isohaline into the estuary in both
configurations, with greater emphasis on low discharge. At low discharge, the maximum larvae
penetration was approximately 109 km from the spawning site to the old (Figure 6A) and
approximately 104 km to the new (Figure 6D) configuration of the Barra Jetties. During the
incidence of the S wind, the larvae were transported up to approximately 102 km in the old



configuration (Figure 6B), and in the new configuration (Figure 6E) up to approximately 101 km.
On the other hand, during SE winds, the larvae were transported up to less than 79 km, reaching
approximately 78 km in both configurations (old and new, Figure 6C and 6F, respectively). During
the incidence of SW winds, the larvae were transported along the central region of the lagoon with
different dispersion patterns. In contrast, during the incidence of the S wind, the larvae were
transported along the west side of the lagoon in the new configuration (Figure 6E) and through the
central cell of the lagoon in the old configuration (Figure 6B). Similarly, during SE winds, the
larvae were transported closer to the west side of the lagoon (Figure 6C and 6F) with little
dispersion.
The total number of larvae transported to the interior of the estuary differed from the old to
the new configuration in each incident south wind (Table 2). At high discharge, the transport was
greater in the old configuration, with more than 6000 (approximately 87%) larvae inside the estuary
in each incident wind (SW, S, and SE). On the other hand, in the new configuration, there was a
reduction in the number of larvae transported according to the incident winds. SW and S winds
carried slightly more than 4000 (~ 61.5%) larvae, differing by approximately 25.5% from the old
configuration, while the SE wind carried only 1287 (~ 18.4%) larvae, a difference of approximately
68.6% from the old configuration. In contrast to the high discharge, in the low continental
discharge, the difference in the transport of larvae to the lagoon between the old and the new
configuration of the jetties fluctuated between 0.5% and 1% according to the winds. SW and SE
winds were the ones that transported the most larvae to the interior of the estuary in both
configurations (old and new), with more than 6000 (~ 86%) larvae each. The S wind carried just
over 5000 (~ 72%) larvae in both configurations.
Table 2: Total number and percentage of larvae of *Micropogonias furnieri* transported towards the
estuary at the end of the 5 days of simulation, during south quadrant winds for the old and new
configurations.

| Wind Direction | Configuration | High Descharge | Low Descharge |
|---|---|---|---|
| SW | Old | 6100 (87%) | 6000 (85%) |
| S | Old | 6016 (86,9%) | 5101 (72,9%) |
| SE | Old | 6101 (87,2%) | 6200 (88,6%) |
| SW | New | 4400 (62,9%) | 6003 (85,8%) |
| S | New | 4210 (60,1%) | 5003 (71,5%) |
| SE | New | 1287 (18,4%) | 6102 (87,2%) |

In the adjacent coastal region, it was observed that some larvae that did not enter the estuary
were transported to the north during SW winds (Figure 5A and 5D, 6A, and 6D), leaving the range
of the area of interest in both configurations. During the S winds (Figure 5B and 5E, 6B and 6E), a
portion of the larvae was trapped at the mouth of the estuary in the coastal region adjacent to the
east of the Barra Jetties. During the incidence of SE winds (Figure 5C and 5F, 6C and 6F), the
larvae were also concentrated in the coastal region adjacent to the mouth of the estuary. In contrast
to the high discharge, in the low discharge, the old and the new configuration did not present
notable differences in the dispersion of the larvae in the coastal region. However, as in the high
discharge, at low discharge, larvae that did not enter the estuary showed a similar dispersion pattern.
**3.3. Larvae Travel Distance**



Figure 7 shows the evolution of the average distance traveled by the particles at the end of
each simulation day for the high (Figure 7A) and low (Figure 7B) discharge period during SW
winds. This wind condition was chosen because it guaranteed the largest incursion of larvae into the
estuary in comparison to the S and SE winds (Figures 5 and 6).
During the high discharge period, at the end of the first day, the eggs covered an average of
13 km in the old and 9.5 km in the new configuration of the Barra Jetties (Figure 7A). On the
second day, the distance covered reached approximately 22.5 km and 19 km in the old and the new
configuration, respectively. On the third day, they passed the central region of the estuary, reaching
35 km and 30.5 km. On the fourth day, the distance covered was reduced; in the old configuration,
the larvae reached approximately 39.5 km, and in the new configuration, they reached
approximately 35.5 km. This trend was maintained, with the fifth day being the shortest route for
the larvae, reaching an average distance of 45 km and 43 km in the old and the new configuration,
respectively. These differences in the mean distance were not statistically significant (p = 0.6857).
At low discharge (Figure 7B), the particles passed the northern limit of the estuary (Ponta de
Feitoria, Figure 1A). On the first day, the particles traveled approximately 17.5 km in the old
configuration and approximately 13.5 km in the new configuration. On the second day, they reached
approximately 31 km and 26 km away, in the central region of the estuary, in the old and new
configuration, respectively. On the third day, they reached approximately 49.5 km and 45 km. On
the fourth day, different from what was observed in the high discharge, the distance covered
increased in both configurations (old and new), and the larvae reached approximately 65.5 km and
60.5 km. This trend was maintained, and on the fifth day, the larvae reached an average distance of
94 km and 89 km in the old and new configurations, respectively. Similar to the high discharge, at
low discharge, the differences in the average distance between the old and the new configuration
were not statistically significant (p = 0.8099).
At the end of each day, the larvae traveled long distances in the old configuration of the
Barra Jetties, at both high and low discharge. At high discharge (Figure 7A), the distances traveled
by the larvae in the two configurations decreased from day 1 to day 5. The difference on the first
day was approximately 3.5 km, gradually reducing to approximately 2 km on the fifth day. The
average distance traveled at the end of each day was similar for both configurations, ranging from
approximately 9.5 km from the first to the second day, increasing by approximately 12.5 km from
the second to the third, and then decreasing to 4.5 km and 5.5 km in the old and new configuration,
respectively. At low discharge (Figure 7B), the average travel difference between the two
configurations was approximately 4 km on the first and second days. On the third day, the
difference between the two configurations increased to 4.5 km, while reducing on the fourth day to
4 km. On the fifth day, the difference was fixed at 5 km.

## 3.4. Larvae Trajectories

To study the evolution of the particle trajectory over time, the transport behavior of 10 larvae
was analyzed from the first hour (1 hour) to the end of 5 days of simulation for both configurations
of the Barra Jetties, during the high and low period (Figure 8) discharge. Experiments with an
incidence of the SW wind were analyzed because they ensured the largest incursion of larvae into
the interior of the Patos Lagoon compared to the S and SE winds.
The largest length of the trajectory traveled by the larvae was observed in the old
configuration both at high (Figure 8A and 8B) and low (Figure 8C and 8D) discharge. During the
period of high continental discharge, most of the tracked larvae entered the estuary in the old
(Figure 8A) configuration compared to the new (Figure 8B) configuration of the Barra Jetties. The
other larvae stayed in the adjacent coastal region (Figures 8A and 8B) and moved to the north. The
greater extension in the daily trajectory traveled by the same larva at the end of each day led the
larvae to position themselves in distant locations between the two configurations, a fact that was
reflected in the final position. Of the larvae that entered the estuary, it was also observed that in the





old configuration of the jetties (Figure 8A), some larvae had their trajectory in the bags of
Mangueira and Arraial (Figure 1A), while in the new configuration of the jetties (Figure 8B), none
of the larvae entered into the bags. At low discharge, the tracked larvae behaved similarly to the
high discharge. The larvae that entered the estuary differed in length by covering greater distances
in the old configuration, and it was also observed that larvae entered into the bags only in the old
configuration of the jetties (Figure 8C).

### 3.5. Spatiotemporal Distribution of Eggs and Larvae

Figure 9 shows the spatiotemporal evolution of eggs and larvae concentration in 6 areas of
the Patos Lagoon estuary (Figure 1C) over the 5 days of simulation during the period of high
continental discharge. For all tested wind scenarios (SW, S, and SE), the number of larvae that
reached regions A1, A2, A3, A4, A5, and A6 differ from each other when comparing results for the
old and the new configuration of the jetties.
During the beginning of the simulation of the high discharge, the two days preceding the
simulations (day 30 and 12/31/2002) presented winds from the south quadrant (Figure 2A, 2C, 2D),
driving the early entry of the plume into the interior of the estuary. One hour after the start of the
simulation, area A1 showed approximately 100% (~ 7000) of eggs in the old configuration and
approximately 61% (~ 4300) of eggs in the new configuration, and no eggs were recorded in the
remaining estuary areas during SW winds (Figure 9A); a similar situation was verified during S and
SE winds (Figure 9G and 9M). At low discharge, one hour after the simulation started, area A1
showed only a concentration of approximately 36% (~ 2500 of 7000) of eggs in the old
configuration, while in the new one, there were no eggs in the estuary for any of the incident winds.
(Figure 10A, 10G and 10M).
On day 1, during the SW wind (Figure 9B), the larvae reached area A3, concentrating the
largest number of larvae in area A2 in both the old and the new configuration. However, different
from the old configuration that already concentrated more than 75% of the larvae, in the new
configuration, the total of the larvae in areas A1, A2 and A3 represented less than 50%. During the
S wind (Figure 9H), in both configurations, the larvae reached area A2 where they concentrated
their greatest abundance of approximately 94% and 84% for the old and new configuration,
respectively. On the other hand, during the SE wind (Figure 9N) at the end of the day1, in the old
configuration, the larvae reached area A2, and almost 100% of the larvae concentrated in these 2
areas. However, in the new configuration, the larvae were restricted to area A1 with only
approximately 26% of the larvae.
On day 2, during the SW wind (Figure 9C), the larvae reached area A4, with the greatest
abundance in area A3 in both configurations; an abundance of approximately 85% in the old
configuration was shown, while that in the new configuration was approximately 64%. During the S
wind (Figure 9I), the larvae reached area A3, registering the greatest abundance in this area, but a
reduction in the total number of larvae in the areas within the estuary was observed compared to
day 1. The old configuration presented approximately 71%, and the new configuration presented
only approximately 15% of the larvae. During the SE wind (Figure 9O), the distribution of the
larvae did not pass from area A2 in both configurations, and similar to the S wind, a reduction in the
total number of larvae inside the estuary was observed, reducing to approximately 54% in the old
configuration and approximately 20% for the new configuration.
On day 3, concentrations increase again in the 3 winds studied (Figures 9D, 9J, 9P). During
the SW wind (Figure 9D), the larvae reached area A5 (northern limit of the estuary, Figure 1C),
with a greater abundance in area A4 in both configurations. A total of approximately 87% (~ 6,100)
of the larvae were recorded in the old configuration and approximately 60% (~ 4,200) in the new
configuration. During the S wind (Figure 9J), the larvae reached area A4 in both configurations, and
the abundance in the old configuration was approximately 86% (~ 6050), and in the new
configuration, the total larvae did not exceed 15% (~ 1030). During the SE wind (Figure 9P), the





larvae reached area A4 in the old configuration where the greatest abundance occurred, while in the
new configuration, area A3 was restricted, and the greatest abundance was observed in area A1. The
total number of larvae was approximately 86% (~ 6060) in the old configuration and approximately
61% (~ 4300) in the new configuration.
On day 4, during the SW wind (Figure 9E), the larvae did not pass area A5, showing only an
increase in abundance in area A5, and the total larvae did not change in both configurations. During
the S wind (Figure 9K), the larvae reached area A5 despite the greater abundance being
concentrated in area A4, as observed on day 3 in both configurations. The total number of larvae
inside the estuary did not change in the old configuration, remaining at approximately 86% (~
6050), while the new configuration registered an increase of approximately 62% (~ 4400). During
the SE wind (Figure 9Q), the larvae did not pass area A4 in both configurations, concentrating
approximately the total of the larvae in this area in the old configuration with an abundance of
approximately 86% (~ 6060). In contrast, in the new configuration, the larvae were distributed in all
areas with a total abundance of approximately 18% (~ 1260).
On the last day of the simulation (day 5), the total number of larvae did not change from day
4 in both configurations for the 3 winds studied. During the SW wind (Figure 9F), the larvae passed
from the northern limit of the estuary (Figure 1C) and reached area A6 in both configurations, with
their greatest abundance in area A5. During the S wind (Figure 9L), the larvae distribution was
limited to area A5 in both configurations, concentrating their abundance between areas A4 and A5.
During the SE wind (Figure 9R), the old configuration showed a decrease in the larvae incursion,
and the greatest abundance was observed in area A3 in the new configuration, where the distribution
of abundance was approximately similar in all areas.
In the low continental discharge, 1 h after starting the simulations for the 3 winds studied
(SW, S, and SE) (Figure 10A, 10G, 10M), only in the old configuration did the larvae enter the
estuary, concentrating the larvae in area A1 with an abundance of approximately 36% (~2500).
On day 1, during the SW wind (Figure 10B), the larvae in the old configuration passed from
the northern limit of the estuary (A5, Figure 1C) and reached area A6 in the old configuration, with
a greater abundance in area A4. In contrast, in the new configuration, the larvae did not pass area
A3 where the greatest abundance was observed. Unlike the old configuration that had the largest
incursion of larvae in the estuary, the new configuration presented the largest number of larvae in
approximately 81% (~5700) compared with approximately 71% (~5000) of the old configuration.
During the S wind (Figure 10H), the larvae reached area A4 in the old configuration, while in the
new configuration, the larvae did not pass area A3 where the greatest abundance was observed in
both configurations. The total abundance was approximately 76% (~5300) in the old configuration
and approximately 24% (~1700) in the new configuration. During the SE wind (Figure 10N), in the
old configuration, the larvae also reached the northern limit of the estuary, area A5, concentrating
their greatest abundance in area A4. In contrast, in the new configuration, the larvae did not pass
area A3, concentrating the greatest abundance in the areas A1 and A2 on the lower estuary. As
observed during the SW wind, in the SE wind, the new configuration presented the highest total
abundance of approximately 92% (~6500) compared to approximately 85% (~ 6100) in the old
configuration despite the lesser incursion.
From day 2 (Figure 10C, 10I, 10O), the larvae in the old configuration passed from the
northern limit of the estuary (area A5, Figure 1C) during all winds (SW, S, and SE), spreading
toward the north of the Lagoon, whereas in the new configuration, this only occurred during the SW
wind. The concentration of larvae declined from region A1 until up to the limit of the estuary (area
A5), with total abundance declining to 57% (~4000) and 47% (~3300) for the old and new
configuration, respectively, concentrating the largest number of larvae in area A4 (Figure 10C) in
both configurations. The S wind showed the same behavior only for the old configuration, where
the abundance declined to approximately 74% (~5150), while in the new configuration, the larvae
were limited and concentrated their greatest abundance in area A4, and their total abundance



registered an increase of approximately 73% (~5100) (Figure 10I). The SE wind showed almost no
larvae from area A4 (Figure 10O) in the new configuration, whereas the old configuration showed
its greatest abundance in area A6. The new configuration presented its abundance distributed in
areas lower than A4 (A1, A2, and A3). Similar to the SW wind, the SE wind showed a decline in
total abundance to approximately 71% (~ 5000) in the old and approximately 86% (~6000) in the
new configuration.
On day 3 (Figure 10D, 10J, 10P), the decay of larvae abundance continued in the region of
the estuary (A1 to area A5) for all winds, demonstrating the incursion of the larvae beyond the
estuarine region. During the SW wind (Figure 10D), in the old configuration, the abundance
dropped to 36% (~2500), and in the new configuration, the larvae reached and passed the northern
limit of the estuary, with their total abundance dropping to 60% (~4200). During the wind S (Figure
10J), the abundance declined to approximately 69% (~4800) and approximately 31% (~2200) in the
old and the new configuration, respectively. However, during SE winds (Figure 10J), both
configurations did not show changes in their total abundance, maintaining approximately 71% (~
5000) in the old and approximately 86% (~6000) in the new configuration, with an advance of
larvae to area A4.
From day 4 (Figure 10E, 10K, 10Q) onward, the new configuration followed the same decay
behavior, similar to the old configuration for the SW and S wind. The southern areas of the estuary
(A1, A2, and A3), showed low and/or almost no larvae and higher concentrations in the northern
areas (A5 and A6) (Figure 10E, 10F, 10K, 10L), demonstrating that part of the larvae may have
crossed the northern limit of area A6 (Figure 1C) because, as illustrated in Figure 6, the maximum
range of the larvae during the simulated winds exceeded 75 km (northern limit of area A6). The
total abundance on the 4th and 5th days decreased to approximately 23% (~1600) in the old
configuration and approximately 59% (~4100) in the new configuration during the SW wind, and
approximately 44% (~3000) for both configurations during S wind. Larvae during the SE wind only
reached the northern limit of the estuary (area A5) on day 5 (Figure 10R) in the new configuration.
The total larvae abundance was approximately 93% (~ 6500) and approximately 90% (~ 6300) in
the old and new configuration, respectively, and the greatest abundance was concentrated in areas
A4 and A5.
## 3.6. Lateral Stratification at the mouth of the estuary
### 3.6.1. Changes on Salinity Stratification
The lateral stratification between the jetties was analyzed only for low discharge
experiments because at the beginning of the experiments, the plume of the Patos Lagoon was in the
adjacent coastal region, allowing the lateral stratification process to be more evident between the
jetties (António et al., submitted). The lateral stratification varied from the old (Figure 11A, 11B,
11C) to the new (Figure 11D, 11E, 11F) configuration for each incident wind direction. The lateral
stratification was more evident during the incidence of the SW wind, both for the old and the new
configuration. The beginning of the flood, and consequently the establishment of lateral
stratification in both the old and in the new configuration of the Barra Jetties, occurred 1 hour after
the beginning of the experiment (Figures 11A and 11D). In the SW wind experiment, the highest
salinity was observed near the east jetty and the lowest near the west jetty, both in the old and the
new configuration. In the new configuration, however, the salinity gradient was smaller, as the
difference in salinity between the east and west was 2.5 psu, while in the old configuration this
difference was 5 psu.
With the S wind experiment, a clear pattern of lateral stratification was not observed in the
two configurations of the Barra Jetties. It should be noted that in the old configuration (Figure 11B)
the lateral stratification between the jetties was not established, but the flow of saltwater was
present between the jetties 7 hours after the beginning of the experiment, dominating the entire
navigation channel. In the new configuration (Figure 11E) at 7 hours, the stratification had not been





established, observing the beginning of the entry of saline water from the coastal region into the
mouth of the jetties. In the old configuration, the highest salinity was observed more centralized,
decreasing for the jetties (Figure 11B). During the SE wind, the lateral stratification was observed
only 10 hours after the beginning of the experiment with the highest salinity in the east jetty (Figure
11C). In the new configuration, 10 hours after the beginning of the experiment, the navigation
channel still did not have lateral stratification (Figure 11F). In general, the beginning of floods and
consequently the establishment of lateral stratification occurred faster in the old than in the new
configuration, resulting in a difference of 2 hours during S winds and 5 hours during SE winds.
### 3.6.2. Larvae Distribution
Analyzing the larvae abundance among the jetties (Figure 12), the dispersion results
corroborate with the salinity gradient (Figure 11). During SW winds (Figure 12A, 12D, 12G),
stratification was observed 1 h after the beginning of the experiment for both configurations. In the
old configuration, the largest number of larvae was concentrated at the root of the jetties, spreading
from the central channel toward the east jetty region (Figure 12A). In the new configuration, despite
stratification also occurring 1 hour after the beginning of the experiment, the largest number of
larvae was concentrated in the center of the jetties, in the east jetty region (Figure 12D). During the
S wind (Figure 12B, 12E, 12H), the stratification occurred after 7 hours of simulation. In the old
configuration, the largest number of larvae was observed in the mouth of jetties in the central
channel region (Figure 12H), concentrating the largest number of larvae in the east jetty and
decaying to the west jetty during transport to the interior of the estuary (Figure 12B and 12E). In the
new configuration, the larvae were observed only in the region of the mouth of the jetties in the
central channel of the jetties (Figure 12H). During the SE wind (Figure 12C, 12F, 12I), the
stratification occurred 10 hours after the beginning of the experiment. In the old configuration, the
largest number of larvae was observed in the mouth region in the central channel of the jetties
(Figure 12I), decreasing linearly from the west to the east jetty during transport to the interior of the
estuary (Figure 12C and 12F). In contrast, in the new configuration, no larvae were recorded
between the jetties 10 h after the beginning of the experiment (Figure 12I).
## 4. DISCUSSION
The present study analyzes the effects of changes in the configuration of coastal structures in
the transport and dispersion of eggs and larvae of the croaker, *Micropogonias furnieri*. The study
analyzed the case of the Barra Jetties on the access channel to the Patos Lagoon in southern Brazil.
Continental discharge and wind are the forces that control circulation in the Patos Lagoon estuary
(Moller et al., 2001; Fernandes et al., 2005; Moller and Fernandes, 2010;Odebrecht et al., 2010).
The analysis considered two extreme discharge conditions, high in 2003 (El Niño) and low in 2012
(La Niña), which reflect the extreme circulation conditions (Moller et al., 2001; Marques and
Möller, 2009) and consequently influence the transport of fish eggs and larvae (Muelbert and Weiss,
1991). Situations with winds from the south quadrant were simulated because they favor the entry
of saltwater (Hartmann and Schettini, 1991; Moller et al., 2001; Moller and Fernandes, 2010;
Marques et al., 2011) and consequently maximize the transport of marine organisms to the estuary
(Muelbert and Weiss, 1991). In each of these situations, only changes in depth and jetty shape were
simulated, which were induced by humans in the project to change the jetties at the entrance to the
Patos Lagoon estuary. The duration of 5 days of the experiments, with a continuous incidence of
south winds, was defined according to the passage of cold fronts in the region (Moller and
Fernandes, 2010), and because it represents the life span of the croaker in which they are passive or
do not yet present active movement (Weiss, 1981).
Differences, both in saltwater intrusion and in the pattern of transport and dispersal of larvae
to the interior of the estuary, were observed between the old and the new configuration of the jetties
in the different scenarios analyzed. According to Dugan et al. (2011), the design and inclusion of





engineering structures in coastal environments alter hydrodynamics, modifying water flow, wave
regime and propagation, sediment dynamics, grain size, and depositional processes. Despite the use
of coastal structures around the world for thousands of years, studies on the physical, environmental
and economic effects of these structures in open and sheltered coastal regions are recent ( Yuk and
Aoki, 2007; Azarmsa et al., 2009; Cunha and Caliari, 2009; Ghashemizadeh and Tajziehchi, 2013;
Lisboa and Fernandes, 2015; Silva et al., 2015; Prumm and Iglesias, 2016; ).
Recent jetty modernization works have reduced the incursion of eggs and larvae into the
estuary in the new configuration compared to the old jetty configuration. The ecological effects
resulting from the construction of these structures have been little studied and understood, and even
less about how they alter the functions and services of these natural ecosystems (NRC, 2007;
Dugan et al., 2011;). The morphology of the Patos Lagoon mouth plays a fundamental role in the
transport and dispersion of eggs (Martins et al., 2007). As a consequence, the larvae took longer to
be transported from one area to another as the incursion into the estuary occurred. In this context,
the transport of fish eggs and larvae to estuarine breeding sites is important to ensure the
recruitment and maintenance of fishing resources (Castro et al. 2005; Vieira et al. 2010;). According
to Robins et al., (2013) the larval transport pattern, whether promoting self-recruitment (retention)
or high connectivity among local populations, is fundamentally important for species that live in
irregular habitats, such as reefs, zones between tides or estuaries.
The differences observed in the pattern and extension of larvae incursion in the Patos
Lagoon estuary may be due to differences in the bathymetry of the access channel, symmetry and
length of the jetties between the two configurations, as well as the convergence and funneling of the
jetties in the new configuration ( Cunha and Caliari, 2009; Lisboa and Fernandes, 2015). As noted
by António et al. (submitted) and Silva et al. (2015), the recent modernization works changed the
hydrodynamics of the estuary, reducing saline intrusion as well as the intensity of flood and ebb
currents by approximately 20%, along the access channel between the jetties. Such factors harm the
flood flows that are responsible for the transport of eggs and larvae in their passive phase to the
interior of the estuary (Castro et al., 2005; Martins et al., 2007; Franzen et al., 2019).
Complementarily, a partial centralization of the flow along the navigation channel occurred
(António et al., Submitted), opposing the previous behavior in which the east jetty was more
dynamic than the west jetty (Cunha and Caliari, 2009), which can cause different residual currents
as observed in the Aransas Pass estuary in the Gulf of Mexico, implying the transport of eggs and
larvae (Brown et al., 2000). Human interventions in estuarine geomorphology lead to changes in the
natural flow of saltwater, leading to the loss of habitat and disturbing the ecocline,(which may be
the limit between freshwater-oligohaline (upper), between mesohaline-mixoeuhalina (medium) and
between euhaline-hyperhaline (lower reaches)) throughout the estuarine system, preventing the fish
from moving between previously connected habitats, especially in the previous ontogenetic phases
(Barletta and Lima, 2019). The results of the study demonstrated that the influence of the local
geomorphology in the variation of circulation conditioned the way that the fish larvae are
transported to the estuary, a crucial parameter when associated with other physical and biological
factors that are determinant for transport during the initial stage of life of the larvae (Able, 2005;
Barletta and Lima, 2019). The factors mentioned also affected the larvae trajectory, and differences
were observed in the estuary between the two configurations. It was also found that at the end of
each day, the average distances traveled were shorter in the new configuration of the jetties. Costa et
al. (2013) analyzed studies carried out in the Mondego estuary and found that in the last decades,
the estuaries have suffered intense anthropic pressure and hydromorphological changes that have
led to a progressive decline in their ecological condition, leading to continuous degradation of the
ecosystem.
The number of larvae that entered the estuary also differed between the old and the new jetty
configuration. The change in the configuration of the jetties resulted in differentiated transport to
the interior of the estuary. The abundance of eggs in the access channel was higher in the old



configuration 1 hour after the beginning of the experiment, a trend that remained from area to area
until the final location of the larvae at the end of the experiment. Combined with the reduction of
flood currents along the navigation channel, the reduction of the cross-sectional area by
approximately 15% (due to the change in the mouth configuration), despite the increase in depth by
approximately 2 m (António et al., submitted; Silva et al., 2015) and the extension of the jetties
after the recent works, may be limiting and/or hindering the incursion of the larvae passively
transported to the estuary. Moreover, increasing or decreasing the current velocities has a direct
impact on the transport of particles and water properties that depend on it (Castro et al., 2005 ;
Cunha and Caliari, 2009). However, in the Patos Lagoon, eggs and larvae of species such as croaker
enter the estuary with the intrusion of saltwater from the adjacent coastal region where the species
spawns ( Castello, 1986; Sinque and Muelbert, 1997; Vieira and Castello 1997; Odebrecht et al.
2017).
The reduced incursion caused by the delay in the entry of eggs and larvae into the estuary in
the new configuration of the jetties will contribute to a greater loss due to their dispersion in the
adjacent coastal region. The reduction in the number of organisms that enter the estuary, which is a
nursery (Odebrecht et al. 2017) and contains more appropriate conditions for development, could
result in negative conditions for the croaker population. Ramos et al. (2006) found that
environmental parameters such as increased river flow can prevent the recruitment of marine
species, leading to a decrease in diversity in the estuarine region as well as variations in abundance,
total diversity, and in the structure of the larval fish assembly, as observed in the Lima estuary. In
the coastal region adjacent to the study, ocean currents, tides, and meteorological conditions are the
main forces responsible for the transport and dispersion of organisms (Acha et al., 2004; Muelbert
et al., 2008). Environments with high tidal energy and high variability in oceanographic conditions,
such as coastal regions, lead to greater dispersion of larvae to places far from the population
(Robins et al., 2013) as well as to areas with conditions that are not appropriate for their
development such as the open sea. Although unfavorable environmental conditions rarely directly
induce larval mortality, they contribute indirectly, prolonging the planktonic phase, so that the
larvae are more exposed to planktonic dispersion, predation, or lack of food (Ellien et al., 2004). In
this sense, hydrodynamics has a fundamental role in the planktonic phase, impacting larval
mortality as found by Ellien et al. (2004), where the weak residual currents in the east of the Sena
Bay contributed to the increase in mortality of *Pectinaria koreni* by reducing the duration of its
larval life stage. Therefore, Araujo et al., (2016) analyzed samples of organisms in Sepetiba Bay for
periods covering 3 decades and found significant differences in the structure of fish communities, as
the number of species, individuals and families have changed and/or decreased in the inner and
intermediate bay. Such findings demonstrate that anthropic changes can lead to a reduction in the
incursion, structure, and composition of organisms in the estuary.
Furthermore, regarding the distance and the quantity, the maximum (final) reach of the
larvae incursion inside the estuary was also reduced in the new configuration of the jetties at both
high and low discharge. In the low discharges, there was a greater incursion and reach of the larvae
to the interior of the estuary, which can minimize losses by ensuring that the larvae enter the estuary
during periods with weak currents (toward the coastal) because the greater water volume in the
estuary, observed normally during El Niño, needs greater flow or time for its outflow, while the
opposite is observed with low flow favoring the input of saltwater into larger areas of the lagoon, a
condition commonly observed in southern Brazil (Acha et al. 1999; Odebrecht et al. 2017). The
differences in the daily distances covered revealed that the maximum (final) distance reached by the
salinity and the larvae at the end of the 5 days was smaller in the new configuration. The reduced
incursion is a response to the effect of the reduced entrance area at the mouth, associated with the
weak currents that transported the larvae along the access channel in the new configuration
compared to the old jetty configuration. A reduced penetration of the organisms will result in the
reduction of the oligohaline limit and in the diversity of organisms in distances previously observed,



which will change the structure and composition of the fauna and the environment inside the
estuaries. This may directly reveal the observation and occurrence of organisms of estuarine-
dependent species further north of the estuary over distances previously recorded. Odebrecht et al.
(2005) found that the reduced limit of the salinity range reduces the oligohaline region of the Patos
Lagoon, an important characteristic for the distribution of species and biodiversity. However, the
reduced distances traveled by the larvae in the new configuration, reflecting the reduced saline
intrusion, may interfere with the number of larvae considering a continuous entry into the estuary,
given that, as argued by Able (2005), a stronger salinity gradient at the estuary-ocean interface may
prove to be a major barrier in terms of quantity for the use of larvae and juvenile fish in these
habitats. Fish wealth and abundance have decreased in the last three decades (1990–2010) in
Sepetiba Bay, and more pronounced changes have been observed in the inner and middle bays,
noting that changes in the salinity gradient have led to spatial changes in fish communities due to
the expansion of the port (Araújo et al., 2017). Such port expansion activities, which included
dredging to deepen the navigation channel, contribute to the degradation of the coast, the
impoverishment of natural habitats, and an increase in the pollutant load in the bay.
The circulation observed among the jetties in the access channel to the Patos Lagoon forces
the establishment of lateral stratification, a pattern that was also observed in the access channel to
the bay of Aransas Pass (Bown et al., 2000; Cunha and Calari, 2009; Marques et al., 2011). The
recent jetty modernization works have also affected the establishment of lateral stratification in
periods of low discharge. A delay in the time of occurrence of lateral stratification was recorded in
the new configuration of the jetties, implying a decrease in the number of eggs and larvae
transported during the incidence of SW and S winds, and the non-entry of eggs and larvae during
the SE wind in the access channel to the estuary. The effect that coincides with the weak flood
currents caused by the change in the configuration of the mouths of the jetties (António et al,
submitted; Silva et al. 2015) is due to the asymmetry in the length of the jetties (Cunha and Caliari,
2009; Moller and Fernandes, 2010). A similar result was found on the jetties at Aransa Pass (Bown
et al., 2000). In contrast to what happened in the bay of Aransas Pass, in the Patos Lagoon, the
asymmetry of the jetties was intense in the old configuration. What justifies the delay in the start
time of the occurrence of lateral stratification, combining symmetry and the increase in the length of
the jetties in the new configuration, which results in the reduction of the intensity of the currents
between the jetties (António et al., Submitted; Silva et al 2015). In this way, it is expected that the
location that holds the highest concentrations of eggs and larvae will be altered as consequence of
the reduction in the daily distance and the maximum extent covered by the eggs and larvae toward
the interior of the estuary. During the lateral stratification between the jetties, each incident wind
presented a specific characteristic in the larvae distribution pattern during the incursion into the
estuary and the location of greater concentration in the new configuration did not change
significantly; however, there is still a need for further analysis since the lateral stratification was not
fully established in the time observed for the old configuration for the 3 winds studied. This fact
meant that the variations in the region with the highest concentration of eggs and larvae observed
were mainly determined by the direction of the incident wind. Dugan et al. (2011) and Moller and
Fernandes (2010) believe that the deepening and narrowing of the tidal channels that result from the
shielding, channeling and construction of jetties and coastal infrastructure have been associated
with the vertical modification of stratification and the increase or decrease in the penetration of
saltwater and hypoxia in urbanized estuaries, a fact that contributes to the destruction of the ecology
of coastal and estuarine ecosystems.
Despite the lack of statistical significance regarding the differences observed between the
old and the new configuration of the Barra Jetties, they suggest a reduction in the entry of eggs and
larvae of the estuarine-dependent organisms, which may aggravate the problem of the decay of
species diversity and abundance. This decay may result in a reduction in the stock of adults and,
consequently, in the fishing stock, which already has losses in the coastal region (Haimovici and





Cardoso, 2017; Odebrecht et al., 2017) since the estuary is the most suitable place for guaranteed retention that enables completing the phases of their life cycle (Acha et al., 1999; Muelbert et al., 2008).

Figure 13 presents the conceptual model of eggs and larvae transport during the incidence of southern winds (SW, S, and SE) for the old and the new configuration of the Barra Jetties. The transport and dispersion of eggs and larvae in the first moments after the start of the simulations are forced by the pattern of velocity currents in the coastal region (Figure 13A, 13B, and 13C), which are determined by the direction of the incident wind. In the experiments in both configurations, re-circulation zones with turns are formed at the root of the west jetty, a place that can concentrate and/ or retain larvae in the central region of the turns, both cyclonic and anti-cyclonic, due to the trapping conditions in the protected region. The coastal circulation forms lines of currents that skirted the jetties which, depending on the wind, favor the transport of organisms to the interior of the estuary. The contribution of these physical conditions conditioned the differentiated transport of eggs and larvae (Figure 13D). The differences in the jetty configurations determined the extension of penetration and abundance of eggs and larvae within the estuary. A shorter incursion time and a greater range and concentration of larvae stood out in the old configuration, while in the new configuration, a delay in stratification time, a reduction of the maximum incursion, and the concentration of larvae were registered.

## 5. CONCLUSION AND REMARKS

The study results demonstrated that the recent modernization works of the Barra Jetties of the Rio Grande affected the extension of recruitment incursion, the abundance, and the distribution of eggs and larvae observed in 2003 (El Niño) and 2012 (La Niña). A reduction in these indices was notable in the new configuration when compared to the old configuration, during the transportation into the estuary with SW, S and SE incident winds. The recent modernization works also changed the time for the start of lateral stratification and how the eggs and larvae enter the estuary, with a delay in the new configuration for the 3 simulated winds (SW, S, SE).

The present study concludes, therefore, that the differences in transport, the dispersion of eggs and larvae to the Patos Lagoon, and the extension and its variability are attributed to hydrodynamic factor changes caused by changes in the geomorphology of the estuarine environment imposed by the jetty configuration changes. However, considering the limitations of the results of the TELEMAC-3D model, and taking into account the complexity of the study in this initial phase of the life cycle of the *Micropogonias furnieri* species, this information serves as a first response to the problem of declining abundance and capture of the adult stock in the Patos Lagoon estuary. Therefore, future work is necessary to continue the investigations and will include studying the biological and behavioral characteristics of the species, such as growth rate, temperature, and mortality, which are fundamental factors for the real dimension of successful recruitment.

## ACKNOWLEDGMENTS

The authors would like to acknowledge CAPES (*Coordenação de Aperfeiçoamento Pessoal de Ensino Superior*) for sponsoring the first author's (MHA) PhD's grant through the *Programa de Pós-Graduação Ciência para o Desenvolvimento* (PGCD), and provided resources support to the Postgraduate Program in Oceanology. We thank the CNPq (*Consejo Nacional de Desenvolvimento Científico and Tecnológico*) for the research grants 308274/2011-3 (EHF) and Proc. 310047/2016-1 (JHM). This study was partially funded by the Brazilian Long-Term Ecological Research Program (PELD) from CNPq (Proc.441492/2016-9) and the Fundação de Amparo à Pesquisa do Estado do Rio Grande do Sul (Proc. 16/2551-0000102-2). We are also grateful to the LOCOSTE (*Laboratório de Oceanografia Costeira e Estuarina)* team for giving support during this research and to Prof. Dr. Osmar Moller for providing the in situ data used to calibration and validation of TELEMAC-3D model.

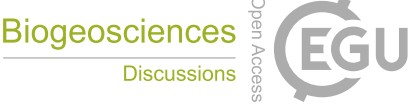

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

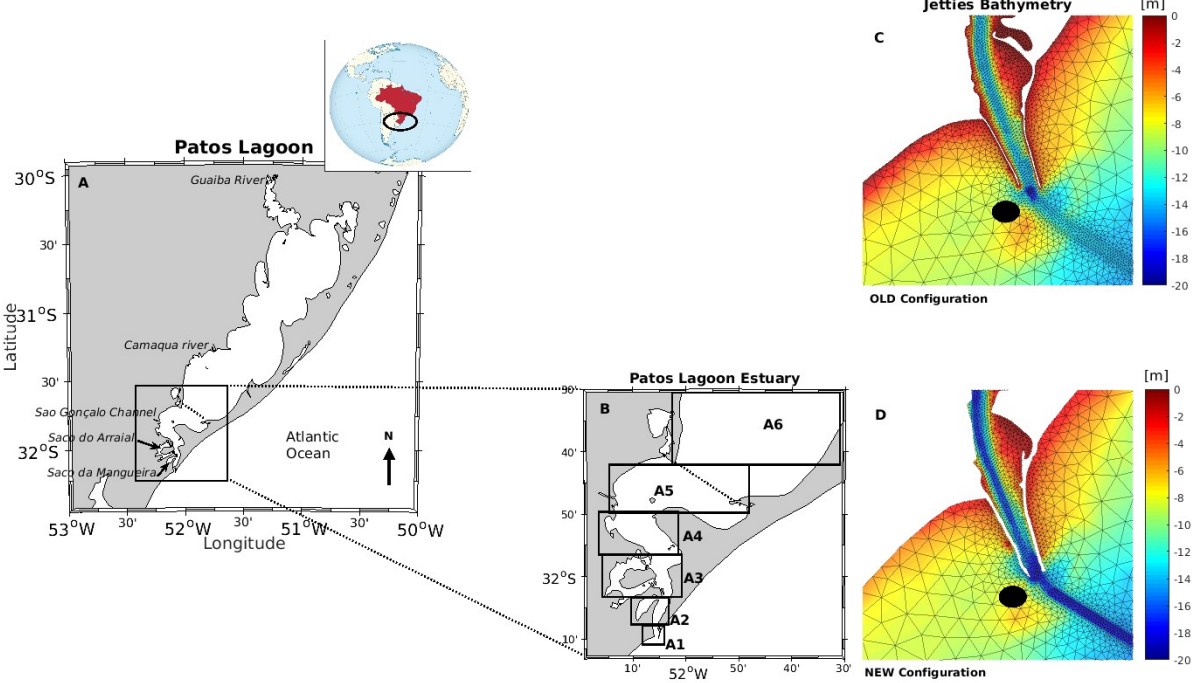

Figure 1: The study site located in Southeast South America, depicting the Patos Lagoon (A) and the estuary (B). Dotted line indicates the estuarine limit (Ponta da Feitoria). Five rectangles areas (A1 – A5) show the location where the model concentration results were extracted. The lower Patos Lagoon estuary in the (C) old, and (D) new jetty configuration. Black points indicate the position where the eggs were released.





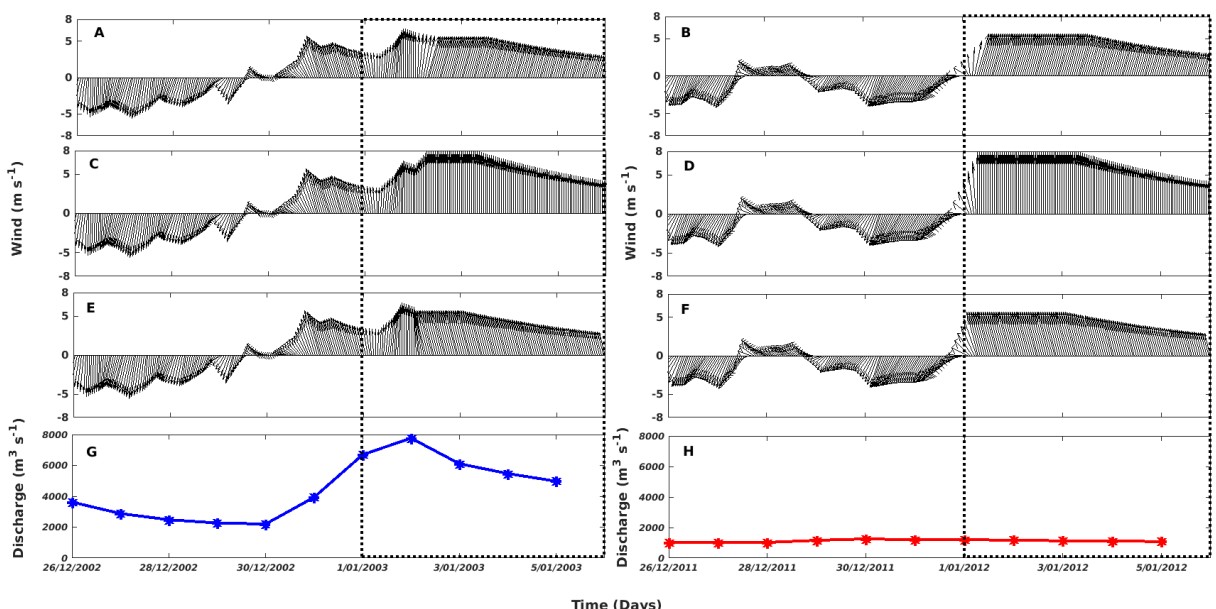

Figure 2: Wind and discharge from 26/12/2002 to 5/01/2003 (left panel) and from 26/12/2011 to 5/01/2012 (right panel). (A and B) SW wind, (C and D) S wind and (E and F) SE wind. Black doted rectangles represent characteristic periods of Patos Lagoon high discharge (G) during El-Niño (left panel) and low discharge (H) during La-Niña (right panel) that were simulated in this study.

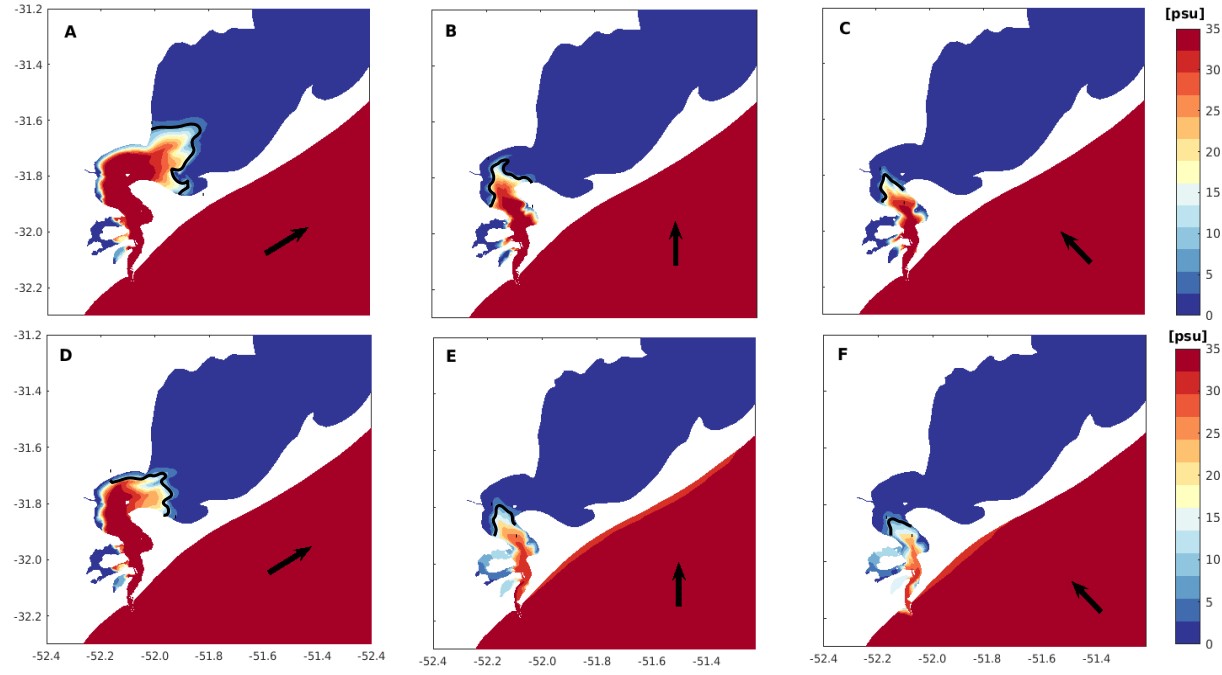





Figure 3: Spatial distribution of salinity excursion at the end of 5 days of simulation during the high
continental discharge condition, considering SW (A and D), S (B and E) and SE (C and F) wind
experiments (black arrows). Results are presented for the old (top panel) and for the new (bottom
panel) jetty configurations. Black line indicates salinity reference of 5 psu.

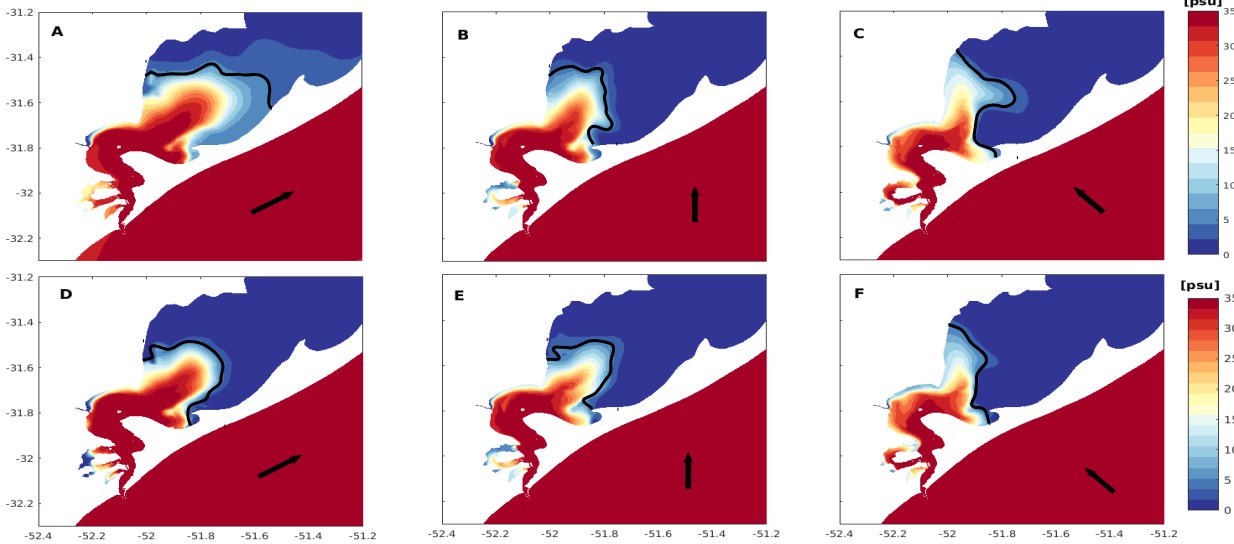

Figure 4: Spatial distribution of salinity excursion at the end of 5 days of simulation during the low
continental discharge condition, considering  SW (A and D), S (B and E) and SE (C and F) wind
experiments (black arrows). Results are presented for the old (top panel) and for the new (bottom
panel) jetty configurations. Black line indicates salinity reference of 5 psu.

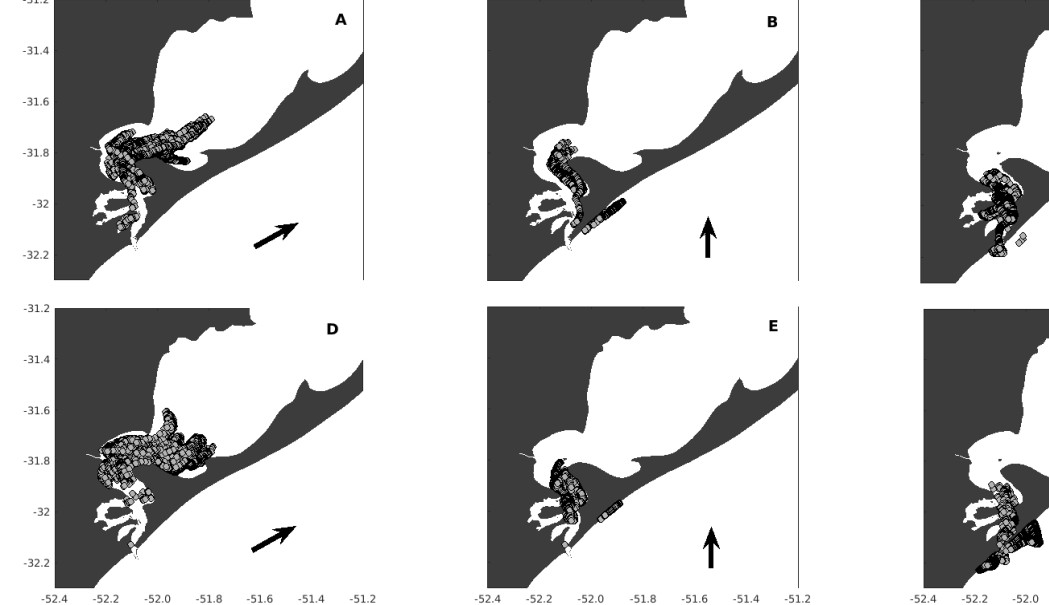





Figure 5: Spatial distribution pattern of excursion of *Micropogonias furnieri* larvae at the end of 5
days of transport, during the period of high continental discharge, considering the SW (A and D), S
(B and E) and SE (C and F) wind experiments. Results are presented for the old (top panel) and for
the new (bottom panel) jetty configurations. Black arrows indicate the wind direction.

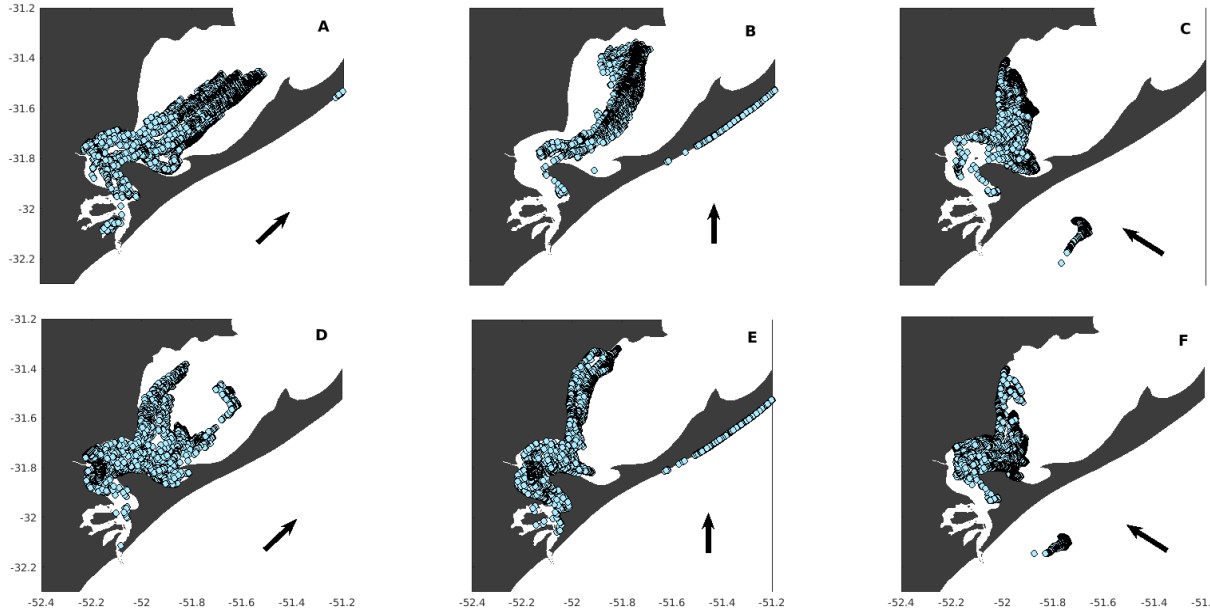

Figure 6: Spatial distribution pattern of excursion of *Micropogonias furnieri* larvae at the end of 5
days of transport, during the period of low continental discharge, considering the SW (A and D), S
(B and E) and SE (C and F) wind experiments. Results are presented for the old (top panel) and for
the new (bottom panel) jetty configurations. Black arrows indicate the wind direction.

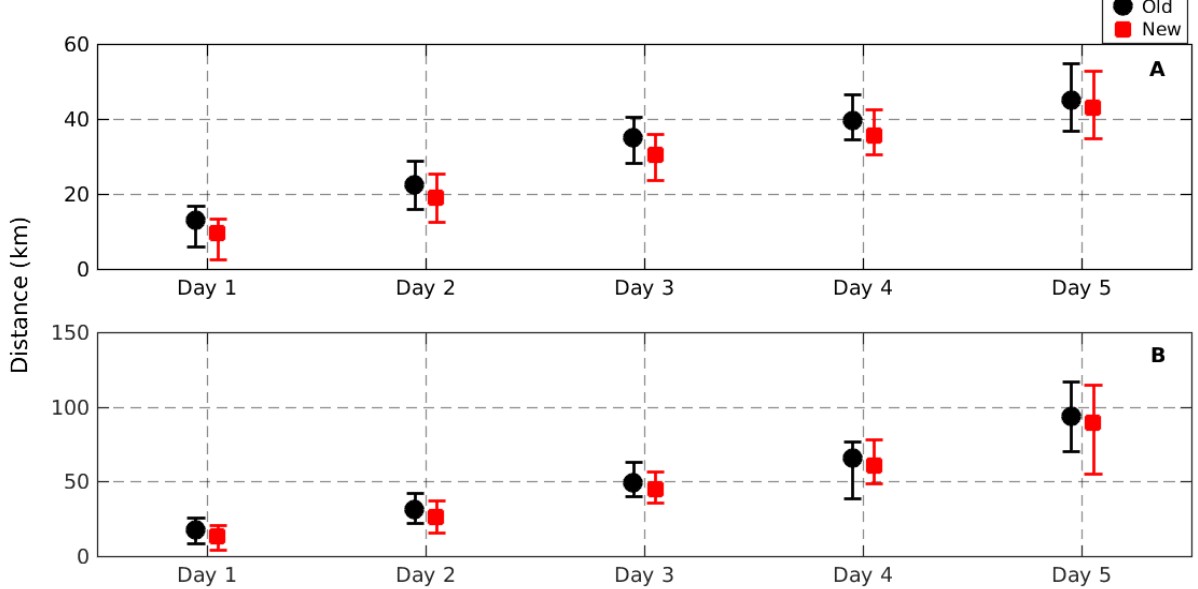

Figure 7: Distribution of the mean distance traveled by *Micropogonias furnieri* larvae sampled
during the SW wind experiment at the end of each of the 5 days (day 1, day 2, day 3, day 4 and day
5), simulated during the period of high water discharge (A) and low water discharge (B), for the old
(black) and the new (red) jetty configuration.

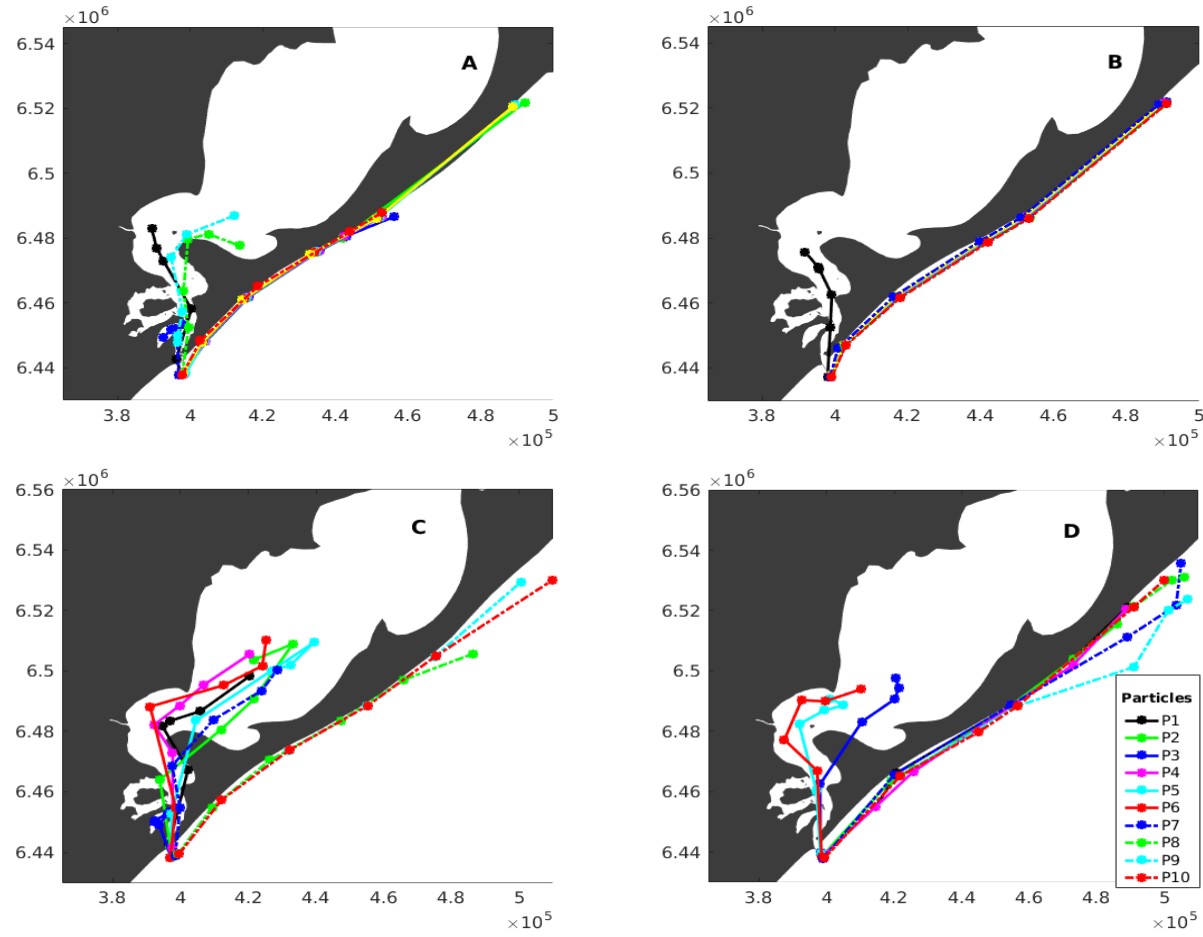

Figure 8: Trajectory of *Micropogonias furnieri* eggs and larvae for the SW wind experiment at the end of each of the 5 days ( 1h, day 1, day 2, day 3, day 4 and  day 5), during the period of high (top panel) and low (bottom panel) water discharge for the old (A,C) and new (B, D) jetty configuration. Particle tracking trajectory during the SW wind experiment, at the end of each of the 5 days of simulation (1h, 1 day, 2 days, 3 days, 4 days and 5 days, marked dots).



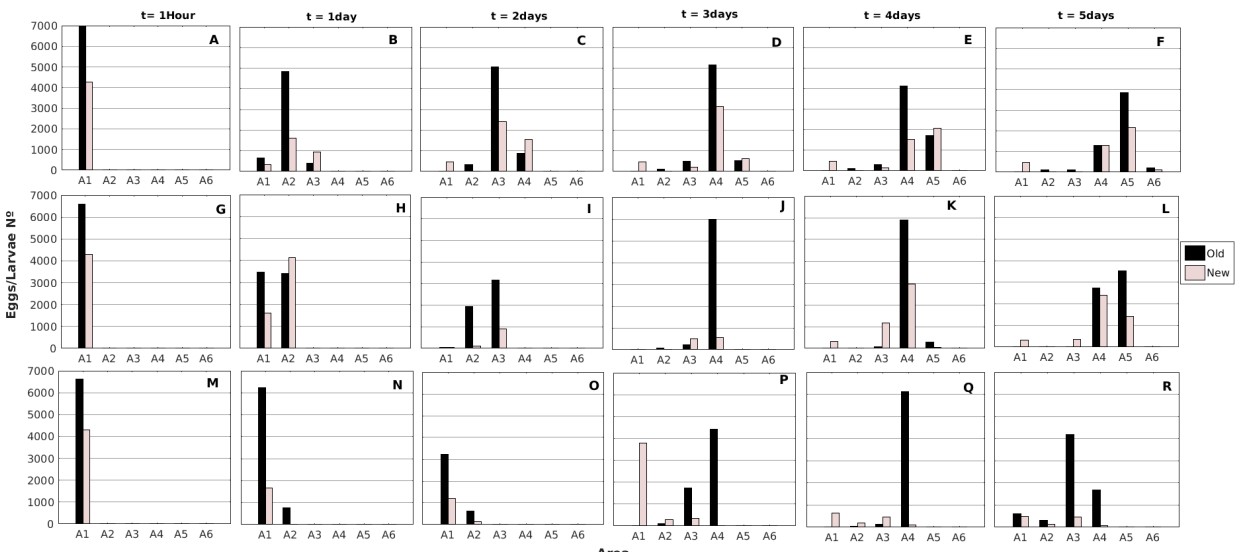

Figure 9: Spatio-temporal distribution of the abundance of eggs and larvae of *Micropogonias furnieri* in the 5 areas (A1, A2, A3, A4 and A5) at the end of each of the 5 days of simulation (day 1, day 2, day 3, day 4 and day 5), for old (blue) and new (yellow) jetty configuration, during the period of high continental discharge. Considering the SW (Top panel), S (center panel) and SE (bottom panel) wind experiments.

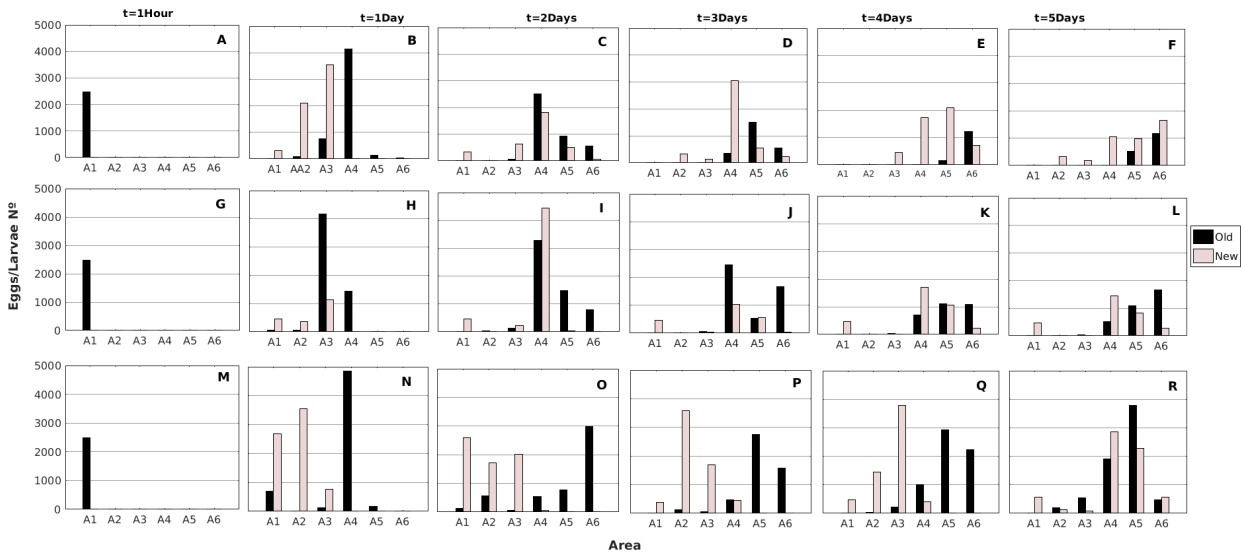

Figure 10: Spatio-temporal distribution the abundance of eggs and larvae of *Micropogonias furnieri* in the 5 areas (A1, A2, A3, A4 AND A5)at the end of each of the 5 days of simulation (day 1, day 2, day 3, day 4 and day5), for old (black) and new (blue) jetty configuration, during the period of low




continental discharge. Considering the SW (Top panel), S (center panel) and SE (bottom panel)
wind experiments.

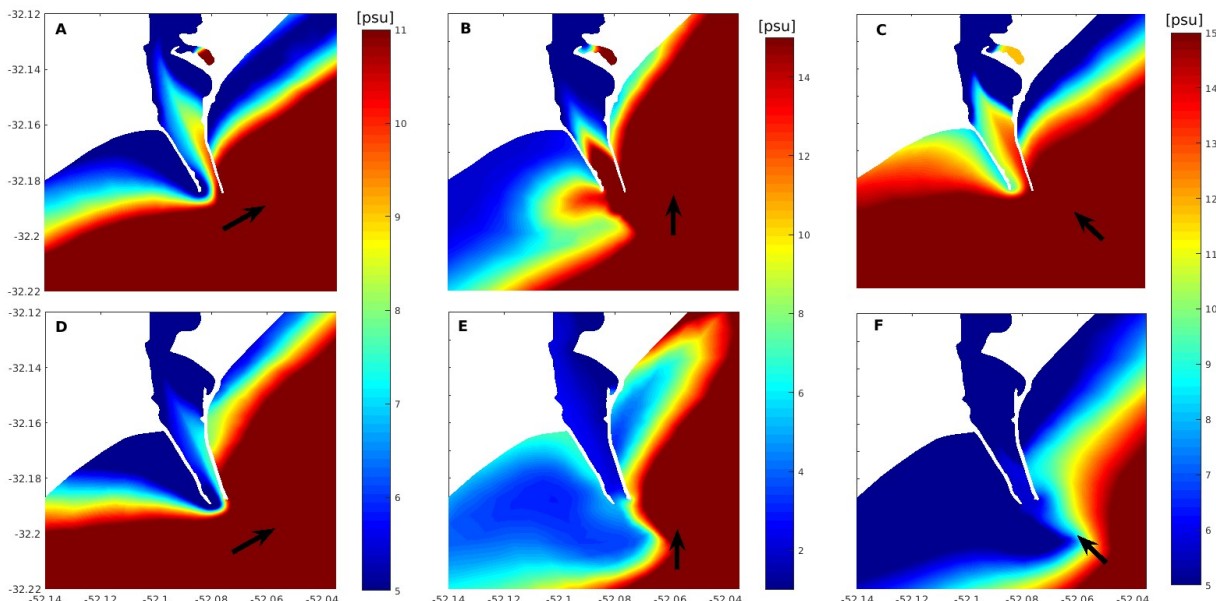

Figure 11: Spatial distribution of salinity  in the estuarine mouth during low continental discharge
at: 1h (A and D), 7h (B and E), 10h (C and F)., considering the SW (A and D), S (B and E) and SE
(C and F) wind experiments. Results are presented for the old (top panel) and for the new (bottom
panel) jetty configurations. Black arrows indicate the wind direction.



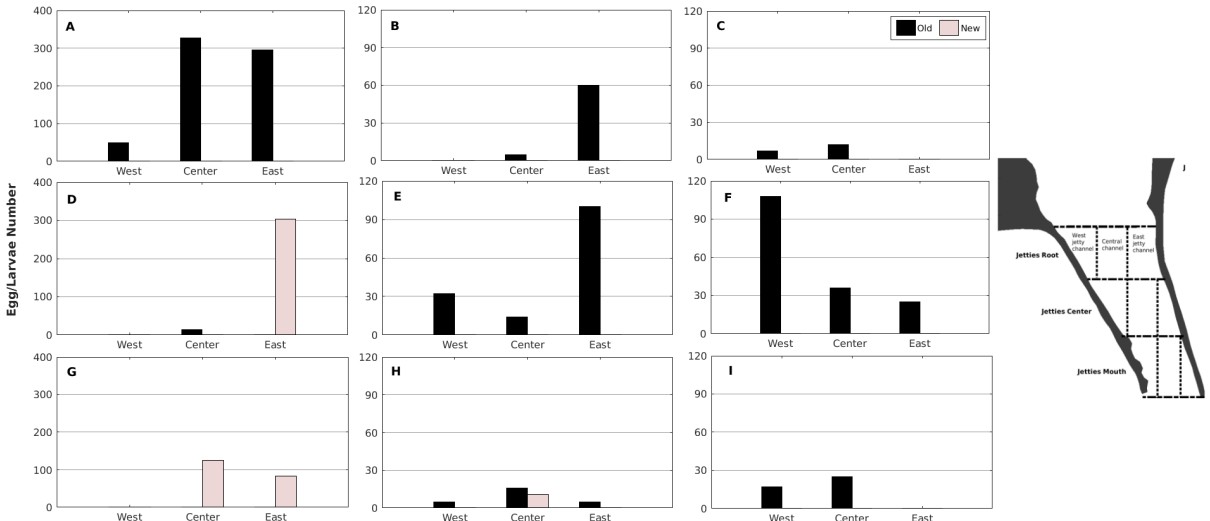

Figure 12: Egg abundance of *Micropogonias furnieri* between the jetties during lateral stratification, considering the SW (A, D and G), S (B, E and H) and SE (C, F and I) wind experiments, at: jetties root (A, B, C), jetties center (D, E, F), an jetties mouth (G, H, I). During the period of low water discharge.

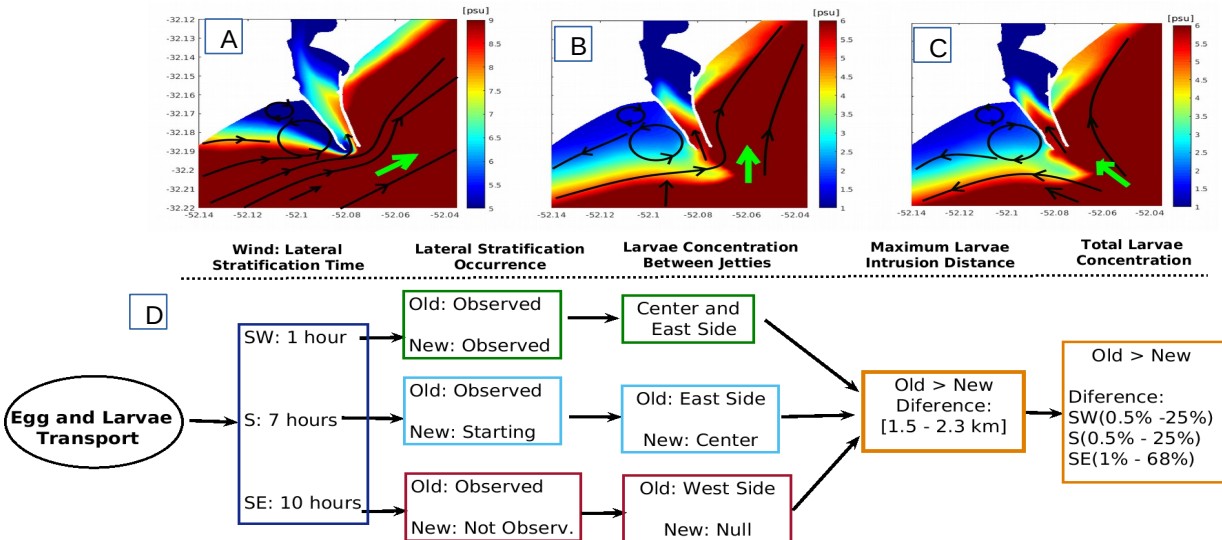

Figure 13: Schematic diagram of the differences in the transport of eggs and larvae of Micropogonias furnieri from the coastal region to the Patos Lagoon estuary induced by changes in the configuration of the jetties.. Coastal circulation induced by SW (A), S (B) e SE (C) winds. Black lines and arrows indicate the current velocity. Green arrows indicate the wind direction. (D) Diagram of the resulted effects in the eggs and larvae transport.