# Peer review of "Human-induced influence on eggs and larval fish transport in a subtropical estuary"

_Biogeosciences, 2020_

## Referee Comment (RC2) · Anonymous Referee #2 · 21 Sep 2020

To evaluate how the Barra Jetties modernization project affects passive transport of fish eggs and larvae into the Plato Lagoon Estuary (PLE), the authors run twelve 5-day experimental simulations using the TELEMAC-3D model with the particle tracking submodule to show the differences in larval dispersal for low versus high river discharge (2 treatments), SW, S, and SE winds (3 treatments), and for the old jetty configuration versus the new jetty configuration (2 treatments). The factorial design of the simulation experiment is good, using 2002-2003 high flow with wind conditions versus 2011-2012 low flow with wind conditions from the field data to define the low and high extremes for the river discharge with the three wind events, and the authors have some good demonstrations of the differences in larval transport and dispersal in the estuary among

the treatments.

However, the writing needs improved before the manuscript is ready for publication, and the manuscript could be shortened considerably to stay more focused on the main objective of the study and the primary results from the modeling simulation exercise to evaluate changes to larval transport based on the new jetties configuration.

The writing for the introduction and the discussion need improved upon and condensed. The biological component of the fish eggs and larvae representing Micropogonias furnieri is not well supported. For example, the eggs/yolk sac larvae could be represented in January by several species in this region besides croaker. Along these same lines, croaker eggs and yolk sac larvae are supplied to the estuary for many more days than 5 in January, so why do these five days necessarily relate to croaker for the simulation exercise? I think you could make the fish eggs/yolk sac larvae general, not mapped to any particular species, especially since the particles are entirely passive with no larval movement behaviors. Then the authors could simply write that that the eggs and yolk sac larvae are passively transported by currents and flow fields, that most egg and yolk-sac larval durations are on the order of 1-5 days, and that this time period chosen in January for the simulation experiment could affect transport of a list of particular species, including M. furnieri, in PLE that are spawned during winter in the coastal waters.

The writing around the simulation experiments is also very broad and not well supported or defined in relation to this particular study. For example, just mentioning other studies that have evaluated larval transport with coastal restoration is not sufficient. The introductory material could be more focused around the objective of the research to evaluate how the configuration of the jetties affects larval ingress and transport into the estuaries.

In the Methods section, I suggest removing lines 282-283. Vertical behavior of larval particles and differences in predator fields have been incorporated into particle trans-
port models, so not necessarily a limitation of the model more so than that the authors didn't do it, correct?

The results section is too long in its current form, with too much description of more results than are necessary, and that are readily apparent within the figures. I suggest that the authors use the figures to describe the overall differences or trends (over days) among the treatments. I think that there are too many figures demonstrating the same overall results that: 1) larval transport into and up the estuary is reduced somewhat by the new jetties configuration compared to the old configuration; 2) larval transport into the PLE is higher under the old jetties configuration than under the new jetties configuration when river discharge is high, with no real difference in transport when river discharge is low; 3) SE and SW winds generally facilitate increased larval transport to the estuaries for both jetty configurations, although not when river flow is high for the new jetty configuration.

For example, I don't think it is necessary to walk through the results for each day in Figures 9 and 10. I suggest deleting the first hour panels and then describe the overall results or trends in larval numbers by section over time between the old and new jetties configuration with references to the winds.

I like Figures 3-6 and offer some minor suggestions below.

Figure 8 is good to show example trajectories for how flow and old vs. new jetty configuration affects larval transport.

Consider removing Figure 7 from the manuscript since Figures 9 and 10 demonstrate the numbers of larvae making it to the six sections in the estuary over days.

The panel labels in Figures 3-6, and then especially for Figures 9 and 10 are okay, but make the figures busier than they need to be. I also think that if the results stay focused on the trends over days and differences among treatments, the alphabetic labels will not be necessary.
Instead, consider adding SW, S, and SE labels with the arrows to the figures and then add "Old Jetties" as top panel label and "New Jetties" as bottom label in Figures 3-6 to more clearly define the treatments in the figure that could also help to describe differences by wind and jetty configuration treatments. Suggest doing the same with Figure 8. Suggest labelling "Old Jetties Configuration" at top of two left panels, "New Jetties Configuration" at top of two right panels, "High Discharge" at right side for top two panels, and "Low Discharge" at right of bottom two panels.

Figures 9 and 10 labelling and as mentioned previously, the panel-by-panel discussion of results is too much. Suggest deleting 1-hour from panels in Figure 9 and 10. I would label top, middle, and bottom panels on right side with SW Wind, S Wind, and SE Wind.

The result that passive particle transport and dispersal confirms or looks similar to the salinity transport results is expected, I think. It seems salinity intrusion into the estuary and the 20% reduction in flood and ebb velocities, is already discussed in Antonio et al. (submitted), so emphasis on salinity changes due to the jetties configuration is not needed.

I suggest the authors try plotting the larval particles in Figure 4 and 5 with the salinity patterns in Figure 2 and 3. It may be too busy and hard to see the salinity gradients through small black dots, but worth a try to show how the salinity and larval transport map together, and to condense the four figures into two figures.

I also suggest removing Section 3.6 and Figures 11-13. Section 3.6 and Figures 11-13 lengthen the paper and add to confusion in describing the results. Although the effects of the new jetties configuration on coastal salinities and flow patterns could be important to larval approach and ingress to the estuary, it seems the larger-scale Figures 2-6, and Figure 8 with larval trajectories, also demonstrate this result to an extent.

For Figures 9 and 10, please explain the differences in numbers by section over days if the larvae don't die? How can the total numbers go up or down over days? Are larvae
transported back out? For example, how can total numbers be approximately 5,000 between section A2 and A3 in panel I in Figure 9, but total numbers be 6,000 the next day in panel J in A4? Where did 1,000 more particles come from the next day?

The discussion restates the results too much and is also too long. The discussion should discuss what the results might mean regarding ingress and transport of fish to the PLE, how the modeling results might be similar or different to other studies and what other studies have shown that support or are different from this study, also maybe discuss model caveats and assumptions and how the modeling exercise could be expanded or improved upon to further evaluate the effects of the jetties on the PLE system and fish resources.

Another discussion point that is mentioned early in the manuscript and then only briefly mentioned in the conclusions is the limitations of the TELEMEC model. The authors do not describe what these limitations are, and why they are important to the modeling exercise? Limitations and caveats to the TELEMEC and larval transport modeling should be part of the discussion. For example, some potential limitations or caveats to cover are why only 5-day simulations, did the authors do more simulations for longer periods of time with continued release of larvae first? Larval fish and eggs are released continuously for much longer time periods than 5 days, why was only passive transport considered? I would think passive particle transport will be somewhat similar to salinity transport, it is the different larval fish behaviors that cause differences in recruitment success and differences from the salinity transport) should be described in the discussion.

Some other comments :

Table 2 is informative. Perhaps try graphing these results using scatter/line plots to show how the results interact among the treatments?

All commas need to be replaced with decimal points such as in Table 2.
Be careful of hanging semicolons in citations like after Prumm and Iglesias 2016 in line 658, and in line 663 after Dugan et al. 2011

For Figure 1: Clarify if panels C and D are scaled to and the insets to the box for A1? Suggest adding lines that point from section A1 to panels C and D if this is the case.

For Figure 2: What specific conditions were used for the 5-day simulations? I think you could either just show the 5-day conditions or decrease size of Figure 2 and scale up the five days to show them as insets on the figure. Figure 2 legend fix to the "dotted boxes around 1/1... to 5/1..." are the periods of time simulated for the 3 wind events and two discharge periods (but again suggest showing what exact conditions were simulated for the 5 days).

---

## Author Comment (AC1) · 24 Sep 2020

Manuscript https://doi.org/10.5194/bg-2020-281-RC1 Response to Reviewer

Maria Helena P. António Instituto de Oceanografia Universidade Federal do Rio Grande Rio Grande, RS – Brazil mhbeula2@gmail.com

Dear Referee 1,

Thank you for your comments and the opportunity of revising our paper on "Human-induced influence on eggs and larval fish transport in a subtropical estuary" for publication in Biogeosciences. We are also grateful for your insightful comments, and the suggestions and comments offered resulted in valuable improvements to our manuscript. Most of the suggestions made were incorporated in the manuscript text. Also, I have included each comment and concern with their respective responses (in bold). We hope this revised version will bring the manuscript to the Biogeosciences standard. We thank you for your valuable contributions.

1. You should clearly state what is new about your work. You cite several past studies that looked at the effect of artificial structures on the marine environment (page 15, lines 654-658); how is your study different and new?

Response: The novelty of this study is to demonstrate that even small alterations in coastal structures can result in circulation changes that have an impact on the transport of planktonic organisms. As discussed in the paper, small changes in fish eggs and larvae transport to nursery areas can lead to recruitment variability. Text inserted in line 137 addresses this issue. (Inserted from line 137).

2. There should be some connection established between the results of your numerical experiments and the real world. The output of the hydrodynamic model has been compared with simulations, but what about the output of the particle-tracking experiments? For example, have there been observations of eggs/larvae or dye experiments anywhere in the world that are consistent with the results of some of your experiments? If so, the results of your other experiments can be seen as something built upon the foundation established by that simulation-observation comparison. Otherwise, this manuscript will be not much more than a report on exercises in running a model.

Response: Thank you for your comment. Our study is the first Lagrangian 3D model with an emphasis on the dispersion of organisms (eggs and larvae), and builds on previous work conducted with 2D models of particle dispersion in the region (Silva et al. 2019, doi: 10.1007/s12665-019-8162-y, Franzen et al. 2019, doi: 10.3856/vol47-issue3-fulltext-15, and Martins et al., 2007, doi: 10.1016/j.jmarsys.2007.02.004). Our

study and previous studies have been used to try to overcome the lack of observational data on the transport of eggs and larvae in the study region. Our literature search did no reveal studies to acquire and to deploy trackable particles with this model that would be suitable for comparison. (Text inserted from line 739 to 767).

3. The description of the hydrodynamic model should be expanded. I realize there is another manuscript that focuses on the hydrodynamic model, but the description here seems to be too brief. For example, which datasets from HYCOM or ECMWF are you using? What are the initial conditions, and how long is the model run to ensure that the model state is produced by the model itself and not just a remnant of the initial conditions? What are the numerical schemes used for the lateral open boundaries, surface momentum flux, advection, mixing, or freshwater flux from rivers? How are the sigma layers distributed vertically?

Response: We have made several changes in the text to address these suggestions (caption 2.1. Hydrodynamic Numerical Model - text below), and a figure with the boundary conditions application scheme has also been included in the manuscript. The model ran for a minimum of 6 months before producing the presented results. "The open boundaries of the domain were forced with results from regional and global models and field data. To be comparable, simulations for both scenarios had the same set-up. Time series of daily averaged river discharge of the main tributaries (Guaíba river and Camaquã river, Figure 1) were obtained from the National Water Agency (www.ana.gov.br) and prescribed at the northern and central continental boundaries. The mean discharge data for the São Gonçalo Channel was considered constant as 700 m3/s (Vaz et al., 2006), as there were no time series of discharge for the studied periods. Temperature and salinity fields obtained from the HYCOM model (Hybrid Model Coordinate Oceanic, https://hycom.org/), with a temporal resolution of 3h and spatial resolution of 1/12.5°, were prescribed tridimensionally in all grid points. Wind time series, with a spatial and temporal resolution of 0.75° and 6h, respectively, were obtained from the ECMWF (European Center for Medium-Range Weather Forecasts,

www.ecmwf.int). Eleven (11) sigma levels were considered in the vertical and distributed from the bottom to the sea surface." (Inserted on line 212).

4. It seems that you are using the Euler scheme to calculate the movement of the particles (page 5, line 230). Is there are reason you are not using a higher-order scheme that are generally known to be more accurate?

Response: The Lagrangian model used is already included in the TELEMAC-3D model and works coupled with the hydrodynamic model. We have opted to use the scheme available in the most updated version of TELEMAC-3D.

5. The quality of the writing needs significant improvement. There are numerous problems, such as vague statements (e.g., "the most important aquatic resources in the world", page 2, lines 47-48), grammatical errors (e.g., "the shallow estuary channel" instead of "a shallow estuary channel", page 2, line 66), misspellings (e.g., "Kjerfev" instead of "Kjerfve", page 4, line 145). The word "salinity" is used when you seem to mean a salinity front (e.g., page 4, line 162; page 8, line 335). The distance traveled by the particles does not seem to be defined in section 3.2 – is it the center of mass or the leading edge of the patch of particles?

Response: The text was corrected and modifications were made accordingly. Your comments and suggestions improved the quality of the paper. Section 3.2 deals with the most advanced edge of the particles. The center of mass is covered in section 3.3.

6. For submitted manuscripts, the convention is for each figure to have its own page, so that readers do not have to flip back and forth between the page with the figure and the page with the caption

Response: Thank you for your comment. A review was made and figures and captions fit to the same page.

7. The list of references has many errors, such as inconsistent formatting, misspellings, and entries out of order. The language in which a source was written should be indi-

cated if other than English. For both the list of references and the authors' information on the first page, please check the journal's policy on whether country names should be in English or can be in the language of that country.

Response: Thank you for your observations. A review was made taking into account the journal's policy to correct these errors.

---

## Author Comment (AC2) · 17 Oct 2020

Response to Reviewer

Maria Helena P. António Instituto de Oceanografia Universidade Federal do Rio Grande Rio Grande, RS – Brazil mhbeula2@gmail.com

Dear Reviewer 2,

Thank you for your comments and the opportunity of revising our paper on "Human-induced influence on eggs and larval fish transport in a subtropical estuary" for publication in Biogeosciences. We are also grateful for your insightful observations, com-

ments, and suggestions because they offered and resulted in valuable improvements to our manuscript. Most of the suggestions were incorporated in the manuscript. Also, I have included each numbered comment and concern with their respective responses. We hope this revised version will bring the manuscript to the Biogeosciences standard. We thank you for your valuable contributions.

1. The writing for the introduction and the discussion need improved upon and condensed. The biological component of the fish eggs and larvae representing Micropogonias furnieri is not well supported.

For example, the eggs/yolk sac larvae could be represented in January by several species in this region besides croaker. Along these same lines, croaker eggs and yolk sac larvae are supplied to the estuary for many more days than 5 in January, so why do these five days necessarily relate to croaker for the simulation exercise?

Thanks for your sugestions, we worked on the text to condense and improved the introduction and discussion. Other species are present in January, but we based our modeling on the croaker since it is the most abundant fish egg and larvae during this time in the environment. The reason for the 5 days simulation is given in line 261 of the Methodology section. The simulation time of 5 days considers the growth rate of the croaker larvae and their passive period in the plankton. Additional information on the early life stage of croaker relevant for the simulation time of 5 days are supplied in that paragraph. Five days is also the average time frame for the passage of cold fronts in the region (line 252).

2. I think you could make the fish eggs/yolk sac larvae general, not mapped to any particular species, especially since the particles are entirely passive with no larval movement behaviors. Then the authors could simply write that that the eggs and yolk sac larvae are passively transported by currents and flow fields, that most egg and yolk-sac larval durations are on the order of 1-5 days, and that this time period chosen in January for the simulation experiment could affect transport of a list of particular species,

including M. furnieri, in PLE that are spawned during winter in the coastal waters.

Thank you for your suggestion. We had initially considered to use a generalized species, since there are other planktonic organisms that are subject to the same passive distribution. However, we constrained the transport experiment to include the growth rate and the duration of the planktonic phase of the croaker (as stated above), and we used known spawning time and location for the species. There is quite a diverse duration of the planktonic phase for fish larvae and the time of 5 days of passive transport would not be universal.

3. The writing around the simulation experiments is also very broad and not well supported or defined in relation to this particular study. For example, just mentioning other studies that have evaluated larval transport with coastal restoration is not sufficient. The introductory material could be more focused around the objective of the research to evaluate how the configuration of the jetties affects larval ingress and transport into the estuaries.

Thank you for your suggestion. We taked them into consideration and review the text accordingly.

4. In the Methods section, I suggest removing lines 282-283. Vertical behavior of larval particles and differences in predator fields have been incorporated into particle transport models, so not necessarily a limitation of the model more so than that the authors didn't do it, correct?

Your consideration is correct. Lines 282-223 were removed.

5. The results section is too long in its current form, with too much description of more results than are necessary, and that are readily apparent within the figures. I suggest that the authors use the figures to describe the overall differences or trends (over days) among the treatments. I think that there are too many figures demonstrating the same overall results that: 1) larval transport into and up the estuary is reduced

somewhat by the new jetties configuration compared to the old configuration; 2) larval transport into the PLE is higher under the old jetties configuration than under the new jetties configuration when river discharge is high, with no real difference in transport when river discharge is low; 3) SE and SW winds generally facilitate increased larval transport to the estuaries for both jetty configurations, although not when river flow is high for the new jetty configuration. For example, I don't think it is necessary to walk through the results for each day in Figures 9 and 10. I suggest deleting the first hour panels and then describe the overall results or trends in larval numbers by section over time between the old and new jetties configuration with references to the winds. Thank you for the proposed observations and suggestions. Figures 9 and 10 and the corresponding description of the results were modified and restructured, focusing on the essential aspects and condensing the information in the results section. The first hour panels in figures 9 e 10 was deleted has suggested.

6. I like Figures 3-6 and offer some minor suggestions below. Figure 8 is good to show example trajectories for how flow and old vs. new jetty configuration affects larval transport. Consider removing Figure 7 from the manuscript since Figures 9 and 10 demonstrate the numbers of larvae making it to the six sections in the estuary over days. The panel labels in Figures 3-6, and then especially for Figures 9 and 10 are okay, but make the figures busier than they need to be. I also think that if the results stay focused on the trends over days and differences among treatments, the alphabetic labels will not be necessary. Thank you for the suggestions to improve the visual aspect of our figures. Figure 7 will be removed and we will focus on the trends over days in Figure 9 and 10.

7. Instead, consider adding SW, S, and SE labels with the arrows to the figures and then add "Old Jetties" as top panel label and "New Jetties" as bottom label in Figures 3-6 to more clearly define the treatments in the figure that could also help to describe differences by wind and jetty configuration treatments. Suggest doing the same with Figure 8. Suggest labelling "Old Jetties Configuration" at top of two left panels, "New

Jetties Configuration" at top of two right panels, "High Discharge" at right side for top two panels, and "Low Discharge" at right of bottom two panels.

These suggestions really improved the figures. Thanks!

8. Figures 9 and 10 labelling and as mentioned previously, the panel-by-panel discussion of results is too much. Suggest deleting 1-hour from panels in Figure 9 and 10. I would label top, middle, and bottom panels on right side with SW Wind, S Wind, and SE Wind. The result that passive particle transport and dispersal confirms or looks similar to the salinity transport results is expected, I think. It seems salinity intrusion into the estuary and the 20% reduction in flood and ebb velocities, is already discussed in Antonio et al. (submitted), so emphasis on salinity changes due to the jetties configuration is not needed.

Figures 9 and 10 were modified as suggested. Text was corrected accordingly to take your suggestions into account.

9. I suggest the authors try plotting the larval particles in Figure 4 and 5 with the salinity patterns in Figure 2 and 3. It may be too busy and hard to see the salinity gradients through small black dots, but worth a try to show how the salinity and larval transport map together, and to condense the four figures into two figures.

It would be very nice to present salinity changes in relation to changes in larval distribution. However, the joint plots became too confusing and we opted to maintain the figures separated.

I also suggest removing Section 3.6 and Figures 11-13. Section 3.6 and Figures 11-13 lengthen the paper and add to confusion in describing the results. Although the effects of the new jetties configuration on coastal salinities and flow patterns could be important to larval approach and ingress to the estuary, it seems the larger-scale Figures 2-6, and Figure 8 with larval trajectories, also demonstrate this result to an extent.

Thank you for your suggestion. We do understand, however, that explicity showing the relation between salinity stratification and larvae distribution is an important issue for this paper and is not clearly presented in the previous Figures. Thus, we decided to keep the paper structure.

10. For Figures 9 and 10, please explain the differences in numbers by section over days if the larvae don't die? How can the total numbers go up or down over days?

Thanks for watching. According to our analysis, the larvae are removed from the estuary due to the increased discharge observed on day 2 ($\sim$ 8000 m3s-1), justifying the decrease in the number of larvae in areas A2 and A3 after their entry on day 1. And as they are transported and pass through the northern limit of the estuary, the concentrations are decreasing in the areas within and within the estuary.

11. Are larvae transported back out? For example, how can total numbers be approximately 5,000 between section A2 and A3 in panel I in Figure 9, but total numbers be 6,000 the next day in panel J in A4? Where did 1,000 more particles come from the next day? Thanks for watching. Figure 2 shows that after increasing the discharge on day 2, on day 3 the discharge decreases, allowing larvae to enter the estuary again.

12. The discussion restates the results too much and is also too long. The discussion should discuss what the results might mean regarding ingress and transport of fish to the PLE, how the modeling results might be similar or different to other studies and what other studies have shown that support or are different from this study, also maybe discuss model caveats and assumptions and how the modeling exercise could be expanded or improved upon to further evaluate the effects of the jetties on the PLE system and fish resources.

We agree the discussion needs improvements and the text was corrected accordingly.

13. Another discussion point that is mentioned early in the manuscript and then only briefly mentioned in the conclusions is the limitations of the TELEMEC model. The

authors do not describe what these limitations are, and why they are important to the modeling exercise? Limitations and caveats to the TELEMEC and larval transport modeling should be part of the discussion. For example, some potential limitations or caveats to cover are why only 5-day simulations, did the authors do more simulations for longer periods of time with continued release of larvae first?

TELEMAC's limitations in the hydrodynamic component were not found. The limitations found were in relation to the number of eggs spawned per cubic meter and not being able to attribute biological characteristics to the particles transported.

The use of 5 days of simulations already done answered in question 1 above. And yes it is possible to do simulations with continuous spawning, but this was not the case in the study.

14. Larval fish and eggs are released continuously for much longer time periods than 5 days, why was only passive transport considered?

Only passive transport was considered because this is the period of time when hydrod-inamic influences dispersion of planktonic organisms. After 5 days, croaker larvae can settle to the bottom and migrate to the shallow embayments where passive transport is not exclusive.

15. I would think passive particle transport will be somewhat similar to salinity transport, it is the different larval fish behaviors that cause differences in recruitment success and differences from the salinity transport) should be described in the discussion.

This issue was addressed in the discussion.

16. Some other comments :

Table 2 is informative. Perhaps try graphing these results using scatter/line plots to show how the results interact among the treatments? All commas need to be replaced with decimal points such as in Table 2. The suggested way of presenting the results allows analyzes that were not possible in the table format.

17. Be careful of hanging semicolons in citations like after Prumm and Iglesias 2016 in line 658, and in line 663 after Dugan et al. 2011

Thank you. The text was corrected accordingly.

18. For Figure 1: Clarify if panels C and D are scaled to and the insets to the box for A1? Suggest adding lines that point from section A1 to panels C and D if this is the case.

Panels C and D are scaled up to represent changes in configuration of jetties and bathymetry. We did draw lines from A1 to panels C and D to represent this.

19. For Figure 2: What specific conditions were used for the 5-day simulations? I think you could either just show the 5-day conditions or decrease size of Figure 2 and scale up the five days to show them as insets on the figure. Figure 2 legend fix to the "dotted boxes around 1/1.. to 5/1. . ." are the periods of time simulated for the 3 wind events and two discharge periods (but again suggest showing what exact conditions were simulated for the 5 days).

You are correct and we agree that legend for figure 2 is not clear regarding the conditions used for the 5 days simulation. We included this information in the figure 2 legend to make it clear: ' Black dotted rectangles represent the 5 days used in the simulation that have characteristic conditions of periods of constant southern wind, wich the intensity decreased linearly from day 2 to day 5.'
* * *